# IgM N-glycosylation correlates with COVID-19 severity and rate of complement deposition

Benjamin S. Haslund-Gourley[1], Kyra Woloszczuk[1], Jintong Hou [1], Jennifer Connors[1], Gina Cusimano[1], Mathew Bell [1], Bhavani Taramangalam[1], Slim Fourati[2], Nathan Mege[1], Mariana Bernui[1], Matthew C. Altman [3], Florian Krammer [4,5,6], Harm van Bakel [4,5,6], IMPACC Network*, Holden T. Maecker [7], Nadine Rouphael [2], Joann Diray-Arce[8], Brian Wigdahl [1], Michele A. Kutzler[1], Charles B. Cairns [1], Elias K. Haddad [1,23] ✉ & Mary Ann Comunale [1,23] ✉

The glycosylation of IgG plays a critical role during human severe acute respiratory syndrome coronavirus 2 (SARS-CoV-2) infection, activating immune cells and inducing cytokine production. However, the role of IgM N-glycosylation has not been studied during human acute viral infection. The analysis of IgM N-glycosylation from healthy controls and hospitalized coronavirus disease 2019 (COVID-19) patients reveals increased high-mannose and sialylation that correlates with COVID-19 severity. These trends are confirmed within SARS-CoV-2-specific immunoglobulin N-glycan profiles. Moreover, the degree of total IgM mannosylation and sialylation correlate significantly with markers of disease severity. We link the changes of IgM N-glycosylation with the expression of Golgi glycosyltransferases. Lastly, we observe antigen-specific IgM antibody-dependent complement deposition is elevated in severe COVID-19 patients and modulated by exoglycosidase digestion. Taken together, this work links the IgM N-glycosylation with COVID-19 severity and highlights the need to understand IgM glycosylation and downstream immune function during human disease.

Severe acute respiratory syndrome coronavirus 2 (SARS-CoV-2), and the disease it causes (coronavirus disease 2019 (COVID-19)), killed more than 14 million people between 2020-21[1]. Once viral particles are inhaled and enter the human airway, the spike (S) protein trimer expressed on the surface of SARS-CoV-2 membranes binds and infects cells via the angiotensin-converting enzyme 2 (ACE2) which is abundant in airway epithelial and endothelial cells[2]. The resulting infection consists of two overlapping phases. The first mainly consists of viral replication associated with mild constitutional symptoms. During the second phase, a combination of the host's adaptive and innate immune response can result in either the efficient clearance of virus-infected cells or the induction of multi-organ system damage

[1]Drexel University/Tower Health Hospital, Philadelphia, PA, USA. [2]Emory University, Atlanta, GA, USA. [3]Benaroya Research Institute, Seattle, WA, USA. [4]Department of Microbiology, Icahn School of Medicine at Mount Sinai, New York, NY, USA. [5]Department of Pathology, Molecular and Cell Based Medicine, Icahn School of Medicine at Mount Sinai, New York, NY, USA. [6]Center for Vaccine Research and Pandemic Preparedness (C-VaRPP), Icahn School of Medicine at Mount Sinai, New York, NY, USA. [7]Stanford University, Stanford, CA, USA. [8]Clinical & Data Coordinating Center (CDCC); Precision Vaccines Program, Boston Children's Hospital, Boston, MA, USA. [23]These authors jointly supervised this work: Elias K. Haddad, Mary Ann Comunale. *A list of authors and their affiliations appears at the end of the paper. ✉e-mail: ee336@drexel.edu; mc375@drexel.edu

requiring intensive care[3]. Patients in this second phase with severe COVID-19 often present with elevated D-dimer[4], C-reactive protein (CRP)[5], IL-6[6], acute kidney injury[7], and heightened complement deposition[8,9].

At the beginning of the pandemic, the Immunophenotyping assessment in a COVID-19 cohort (IMPACC) study was designed as a prospective longitudinal study. Hospitalized COVID-19 patients were enrolled from May 2020 to March 2021, and detailed clinical, laboratory, and radiologic data were collected[10,11]. Biological samples including blood, nasal swabs, and endotracheal aspirates were collected at multiple time points during hospitalization. Patient trajectories were defined by severity of illness over the first 28 days. These patient trajectories were divided into 5 groups based on longitudinal observation of ordinal scores reflecting the degrees of respiratory illness and the presence or absence of complications at discharge[12]. Trajectory Group 1 was characterized by a brief hospital stay without major complications. Trajectory 2 had an intermediate length of stay with no complications upon discharge. Trajectory 3 was characterized by an intermediate length of stay with limitations at discharge. The most severe trajectory groups were 4 and 5. Trajectory 4 had a longer length of stay (~28 days) with complications, while Trajectory 5 was characterized by fatal illness by day 28. Thus, the curation and stratification of these samples provided an opportunity to determine how human IgM glycosylation relates to acute SARS-CoV-2 infection severity.

The glycosylation of immunoglobulins plays an important role during the adaptive immune response to infection and vaccination[13–16]. IgG is the best example of how variations in immunoglobulin glycosylation modulate downstream immune responses. The size and charge of IgG N-glycans occupying Asn-297 site of the Fc heavy chain can promote antibody-dependent cellular-cytotoxicity (ADCC), antibody-dependent cellular phagocytosis (ADCP), Fc-gamma receptor affinity[17–22], and complement activation[21,23,24]. In hospitalized COVID-19 patients, the sialic acid and galactose content on total IgG N-glycans was reduced compared to patients with mild cases of COVID-19 and healthy controls[25]. Furthermore, anti-spike IgG isolated from hospitalized COVID-19 patients contained lowered core-fucose levels in severe patients[26–31], promoting macrophage release of IL-6 and TNF-α and the destruction of endothelial barriers in vitro by binding FcγR IIA and IIIA[32].

While much attention has been paid to the glycosylation of IgG, less has been focused on IgM. IgM is the third most abundant circulating immunoglobulin and is produced early during the adaptive immune response to SARS-CoV-2 infection[33]. Moreover, IgM is a potent immune protein. A single immune-complexed IgM can initiate the complement cascade[34] and plays important roles during early immune responses to clear bacteria, viruses, parasites, apoptotic cells, and is likely involved in promoting immune tolerance[35]. The heavy chain of IgM contains five separate N-glycosylation sites harboring complex-type, hybrid, and highly-mannosylated N-glycans[36,37]. Complex type N-glycans populate IgM at Asn-171, Asn-332, and Asn-395 while Asn-402 and Asn-563, located closer to the tail of the IgM heavy chain, are populated with mannosylated N-glycans[37–39]. In cell lines treated with tunicamycin to block glycosylation of IgM, secretion of IgM was reduced by >95%[40], demonstrating N-glycan's crucial role in the secretion of IgM from B-cells. In vivo, increased IgM sialylation was associated with heightened T cell inhibition[41]. In addition, evidence supports IgM N-glycans interacting with C1q in the classical complement pathway[42] and the mannan-binding lectin (MBL) associated with the lectin pathway of complement activation[38,43]. The recently discovered IgM-specific receptor, FcμR, expressed on NK, B, and T cells has implicated IgM in controlling cellular activation and antibody production[44]. Additional receptors for IgM Fc include Fcα/μR expressed by germinal center follicular dendritic cells[45] and polymeric Ig receptor (pIgR) requiring the J-chain pentamer of IgM for transcytosis

to mucosal surfaces[46]. However, the function of IgM N-glycans interacting with these receptors remains to be explored.

While the N-glycosylation of IgM has been characterized previously in healthy pooled human serum, during cancer[38,47–49], and in recombinant IgM[37,50], this is the first characterization of the IgM N-glycosylation profile isolated from humans with an acute viral infection. Here, we report significant differences in the total and antigen-specific IgM N-glycan content from cohorts of hospitalized COVID-19 patients separated by severity, contrast these findings with patient's IgG N-glycans, and identify potential functional consequences of IgM N-glycosylation.

## Results

### IgM di-sialylation and mannosylation associate with COVID-19 severity

Plasma from patients admitted to the hospital after testing positive for SARS-CoV-2 was analyzed 4- and 7-days following hospital admission. Clinical characteristics of the patients are presented in Table 1 stratified by trajectory 1-5, with 1 being a mild SARS-CoV-2 infection and 5 being death from complications of SARS-CoV-2 infection within 28 days. N-glycan profiles isolated from purified total IgM were analyzed (Fig. 1a), with N-glycan identities listed in Supplemental Table 1. N-glycans ranging from mono-antennary to tri-antennary as well as hybrid and mannosylated moieties were observed in all IgM samples. The 36 individual IgM N-glycan peaks with identities confirmed by mass-spectrometry from day 4 and day 7 are included in Supplemental Figs. 1 and 2. To analyze general trends in the IgM N-glycan profile across disease severity, glycans were grouped by size, charge, and type into classes (G0, G1, G2, S1, etc.) as denoted below the IgM N-glycan profile in Fig. 1a.

Protein glycosylation is impacted by factors including sex, age, and body mass index (BMI)[51–62]. Therefore, COVID-19 patient cohorts from the IMPACC study were analyzed to determine if there were statistically significant differences between mild (trajectories 1 and 2), moderate (trajectory 3), and severe (trajectories 4 and 5) (Fig. 1b). There was no statistically significant difference between cohorts based on sex, age, BMI, the number of days of COVID-19 symptoms prior to hospitalization, or viral load. Furthermore, we determined that there was no statistically significant difference in the concentration of total IgM isolated between each patient cohort (Supplemental Fig. 3). After confirming that cohort characteristics were comparable, we analyzed the IgM N-glycosylation profiles from day 4 and 7 hospitalized COVID-19 IMPACC patients across illness severity (Fig. 1c). Di-sialylated (S2) N-glycans on IgM increased significantly in the severe COVID-19 cohort on day 4 of hospitalization compared to the mild and moderate cohorts. In addition, total mannose, including hybrid N-glycans, decreased significantly in the severe COVID-19 cohort on day 4 IgM. On day 7, the severe cohort's IgM N-glycosylation maintained the trends observed on day 4, but lost significance likely due to the death of four of the COVID-19 patients in the severe trajectories reducing the power of the analysis. Taken together, the changes in IgM N-glycosylation correlate with the severity of SARS-CoV-2 infection in humans.

### IgG and IgM N-glycans responses differ during COVID-19

We next compared the glycosylation of total IgM and IgG isolated from COVID-19 patients to characterize the general plasmablast glycosylation response to viral infection. Patients were sorted into nonsevere (trajectories 1-3) and severe (trajectories 4 and 5) cohorts to compare the change in immunoglobulin N-glycosylation by glycan class. First, IgG N-glycans from healthy control, nonsevere, and severe COVID-19 cohorts were analyzed as grouped classes as described in Supplemental Fig. 4. IgG in both severe and nonsevere COVID-19 exhibited reduced di-galactosylation (G2) and mono-sialylation (S1) while agalactosylation (G0) significantly increased compared to healthy controls in the severe COVID-19 cohort (Fig. 2a). Interestingly, the IgG

**Table 1 | Drexel IMPACC Cohort**

| Trajectory (n) | | total (n = 22) | 1 (n = 1) | 2 (n = 5) | 3 (n = 6) | 4 (n = 2) | 5 (n = 8) | 1–3 Nonsevere (n = 12) | 4–5 Severe (n = 10) |
|---|---|---|---|---|---|---|---|---|---|
| Age | Mean Years (+/- S.D) | 61.6 (15.5) | 34 | 55 (17.4) | 64.5 (14.7) | 65.5 (19.1) | 66 (13) | 58 (17) | 65.9 (13.1) |
| Sex | # Male (%) | 13 (60) | 1 (100) | 3 (60) | 3 (50) | 1 (50) | 5 (63) | 7 (58) | 6 (60) |
| Race | White | 19 (86) | 0 | 5 (100) | 5 (83) | 1 (50) | 8 (100) | 10 (83) | 9 (90) |
| | Black | 1 (5) | 0 | 0 | 0 | 1 (50) | 0 | 0 | 1 (10) |
| | Unknown | 2 (9) | 1 (100) | 0 | 1 (17) | 0 | 0 | 2 (17) | 0 |
| Ethnicity, No. (%) | Non-Hispanic | 15 (68) | 1 (100) | 2 (40) | 5 (83) | 1 (50) | 6 (75) | 8 (67) | 7 (70) |
| | Hispanic | 7 (32) | 0 | 3 (60) | 1 (17) | 1 (50) | 2 (25) | 4 (34) | 3 (30) |
| Comorbidities, No. (%) | None | 5 (23) | 1 (100) | 2 (40) | 1 (17) | 0 | 1 (13) | 4 (34) | 1 (10) |
| | Hypertension | 10 (45) | 0 | 1 (20) | 4 (67) | 1 (50) | 4 (50) | 5 (42) | 5 (50) |
| | Diabetes | 8 (36) | 0 | 1 (20) | 2 (33) | 1 (50) | 4 (50) | 3 (25) | 5 (50) |
| | Chronic Lung Disease | 5 (23) | 0 | 2 (40) | 0 | 1 (50) | 2 (25) | 2 (17) | 3 (30) |
| | Asthma | 4 (18) | 0 | 2 (40) | 1 (17) | 0 | 1 (13) | 3 (25) | 1 (10) |
| | Chronic Cardiac Disease | 4 (18) | 0 | 0 | 0 | 1 (50) | 3 (38) | 0 | 4 (40) |
| | Chronic Kidney Disease | 3 (14) | 0 | 0 | 1 (17) | 0 | 2 (25) | 1 (8) | 2 (20) |
| | Chronic Neurological Disorder | 1 (5) | 0 | 0 | 0 | 1 (50) | 0 | 0 | 1 (10) |
| | Autoimmune Disease | 3 (14) | 0 | 1 (20) | 0 | 0 | 2 (25) | 1 (8) | 2 (20) |
| | Malignancy | 1 (5) | 0 | 0 | 0 | 0 | 1 (13) | 0 | 1 (10) |
| | Smoking (Current) | 4 (18) | 0 | 0 | 0 | 0 | 4 (50) | 0 | 4 (40) |
| | Smoking (Former) | 6 (27) | 0 | 2 (40) | 2 (33) | 1 (50) | 1 (13) | 4 (34) | 2 (20) |
| Body Mass Index (BMI) | Overweight (25.1-29.9) | 7 (32) | 1 (100) | 1 (20) | 1 (17) | 0 | 4 (50) | 3 (25) | 4 (40) |
| | Class I (30-39.9) | 10 (45) | 0 | 4 (80) | 4 (67) | 1 (50) | 1 (13) | 8 (67) | 2 (20) |
| | Class III (40+) | 5 (23) | 0 | 0 | 1 (17) | 1 (50 | 3 (38) | 1 (8) | 4 (40) |
| Symptom Onset to Hospitalization | <3 days | 8 (36) | 0 | 3 (60) | 0 | 1 (50) | 4 (50) | 3 (25) | 5 (50) |
| | 4 to 7 days | 4 (18) | 0 | 0 | 3 (50) | 0 | 1 (13) | 3 (25) | 1 (10) |
| | 8 to 14 days | 4 (18) | 1 (100) | 1 (20) | 1 (17) | 0 | 1 (13) | 3 (25) | 1 (10) |
| | >14 days | 1 (5) | 0 | 0 | 0 | 0 | 1 (13) | 0 | 1 (10) |
| Level of respiratory support (%) | Mechanically ventilated | 4 (18) | 0 | 0 | 0 | 2 (100) | 2 (25) | 0 | 4 (40) |
| | High Flow Nasal O2 | 3 (14) | 0 | 0 | 2 (33) | 0 | 1 (13) | 2 (17) | 1 (10) |
| | Supplemental Oxygen | 10 (45) | 0 | 3 (60) | 4 (67) | 0 | 3 (38) | 7 (58) | 3 (30) |
| SOFA Score | Mean (+/- S.D.) | 2 (3.9) | 0 | 0.2 (0.4) | 0.3 (0.8) | 10.5 (2.1) | 2.6 (0.6) | 0.3 (0.6) | 4.2 (5.1) |
| Remdesivir | #, (%) | 7 (32) | 1 (100) | 2 (40) | 2 (33) | 1 (50) | 1 (13) | 5 (42) | 2 (10) |
| D-Dimer | Mean (+/- S.D.) | 3 (4.2) | 0.8 | 0.7 (0.3) | 1 (0.4) | 12.7 (0.9) | 3.2 (3.6) | 0.9 (0.4) | 5.3 (5.2) |
| BUN | Mean (+/- S.D.) | 26.7 (16.3) | 11 | 14 (9) | 18 (7.4) | 50 (2.8) | 36.3 (14.7) | 15.5 (7.8) | 39 (14.2) |
| Creatinine | Mean (+/- S.D.) | 1.1 (0.5) | 1 | 0.8 (0.2) | 0.8 (0.2) | 2 (0.4) | 1.3 (0.5) | 0.8 (0.2) | 1.4 (0.6) |

Hospitalized COVID-19 patient plasma cohorts are sorted into trajectories 1-5 based on symptom severity assessed by the IMPACC study definition with demographic and clinical parameters, and into cohorts of nonsevere (trajectory 1, 2, and 3) and severe (trajectory 4 and 5).

N-glycosylation of the severe and nonsevere cohorts exhibited significant differences for the G2 and S1 classes. In contrast, the IgM N-glycosylation from the same patients revealed multiple significant changes between severe and nonsevere cohort N-glycan classes (Fig. 2b). G0 and mono-galactosylated (G1) N-glycans significantly decreased in severe patients compared to the nonsevere cohort. Further, the increase in S2 remained significant while tri-sialylated (S3) content also increased significantly in the severe COVID-19 cohort. In comparison, the S2 sialylation of severe patient IgG N-glycans remained lowered or unchanged on day 4 compared to healthy controls (Fig. 2a), aligning with previous studies of IgG N-glycosylation in hospitalized COVID-19 patients[26,29,30]. Lastly, the decrease in mannose remained significant in severe trajectory patients compared to nonsevere patients on day 4 of hospitalization.

The decrease in total mannose content required further interrogation because 11 hybrid and mannosylated N-glycans contribute to the overall decrease observed in IgM during severe COVID-19 (Fig. 2c). The decrease in total mannose was predominantly due to

lowered levels of the smaller hybrid moieties: M4G1, FM4A1, and M5A1 in combination with the mannosylated moieties: M5 and the two isoforms of M6. Mannosylated structures or co-eluting peaks larger than M6 did not significantly decrease, while M9 significantly increased in the severe COVID-19 cohort. Next, mannose and hybrid structures ranging from M4-M6 were compared to mannose structures M7-M10, revealing a potential reduction in the degree of mannose processing by Golgi-bound mannosidases during IgM production. Taken together, the glycosylation pattern of IgM was consistently altered in the severe COVID-19 cohort, with major classes of IgM N-glycans trending in opposite directions compared to the IgG N-glycan classes.

**Glycosyltransferase expression correlates with IgM N-glycosylation**

The observed changes in IgM N-glycosylation likely result from glycosyltransferase expression within the Golgi of B cells or plasmablasts. While plasmablast-specific transcriptomics were not available,

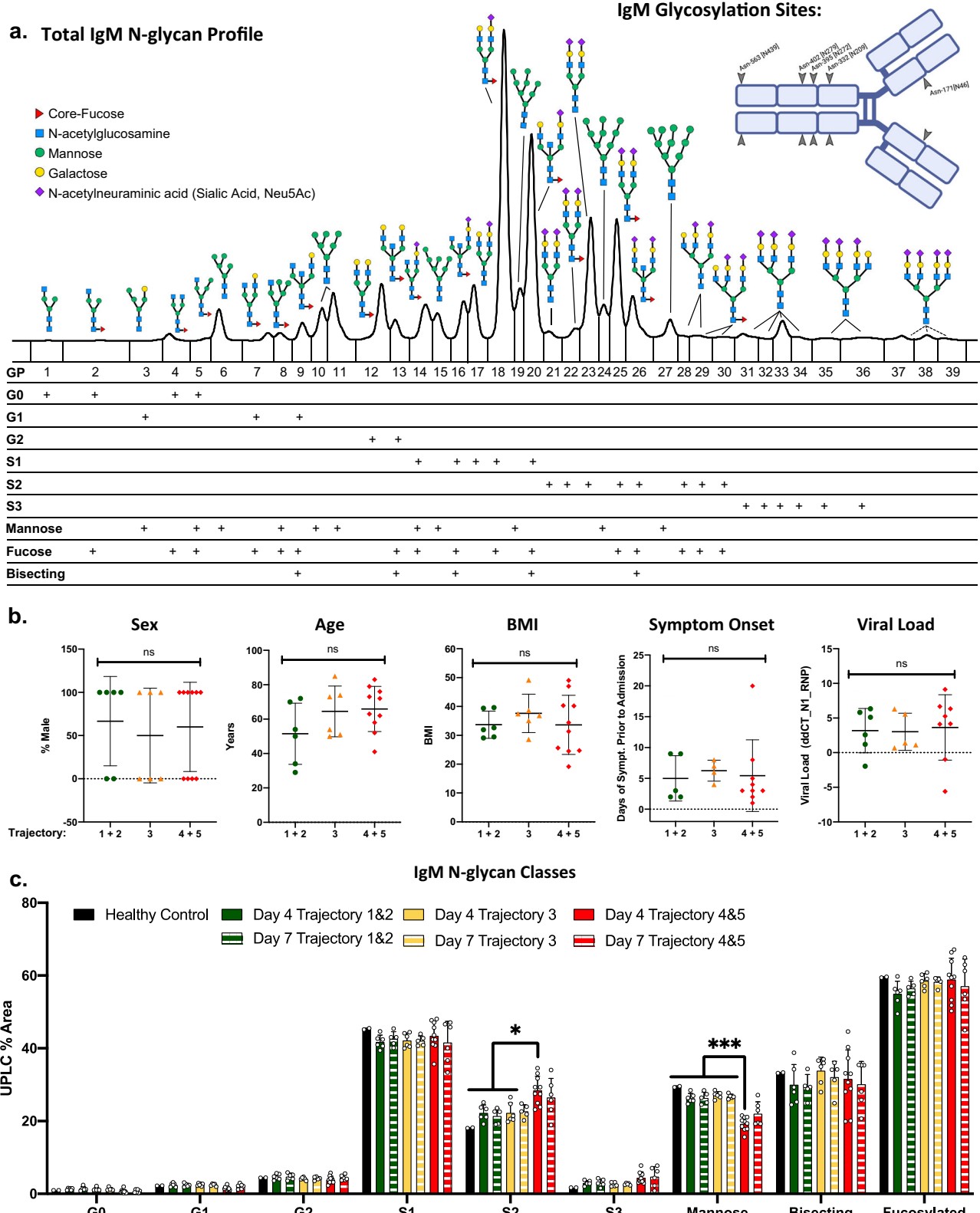

IMPACC study collaborators at Emory University provided glycosyl-transferase and glycosidase transcript expression data isolated from peripheral blood mononuclear cells (PBMCs) collected on day 0 of patient hospitalization. After normalizing the data by total read count and transforming by log2 for comparability, expression profiles were compared between the severe and nonsevere COVID-19 trajectories.

The expression of the mannosidases MAN1A2 and MAN2A1 decreased significantly in the severe cohort compared to the non-severe cohort (Fig. 3a). These mannosidases are responsible for processing high mannose structures into smaller mannose moieties[63]. The decrease in mannosidase expression aligns with data in Fig. 2c where we observe less mannosidase-processed M5 and M6 content in the severe COVID-19 cohort IgM. In addition, IgM total mannose

**Fig. 1 | IgM N-glycosylation analysis reveals differences in COVID-19 patients stratified by trajectory. a** IgM N-glycans labeled with the RapiFluor (RFMS) were profiled with UPLC-FLR-ESI-MS. The resulting N-glycans were identified using mass spectrometry and retention time data. Please see Supplementary Table 1 for a complete list of N-glycans. Dashed lines represent N-glycans without confirmed mass identities due to the limitation of the RFMS label in the QDa mass spectrometer. IgM monomer is displayed with the 5 conserved glycosylation sites labeled, created using BioRender. GP = glycan peaks. **b** Cohort demographics: Sex, age, body mass index (BMI), time from symptom onset to hospital admission, and viral load expressed as the delta-delta change between SARS-CoV-2 nucleocasid protein 1 (N1) and the house keeping gene RNP via RT qPCR are presented stratified across trajectory 1–2 ($n = 6$), 3 ($n = 6$), and 4-5 ($n = 10$) data are presented at mean values +/– S.D. Data was analyzed for significance using a one-way ANOVA with Tukey's multiple comparisons test. **c** IgM N-glycans are grouped by class: G0 refers to core diantennary N-glycans lacking galactose, G1 refers to core diantennary N-glycans with a single galactose, G2 refers to core diantennary N-glycans with two galactoses, S1 refers to diantennary N-glycans with a single sialic acid, S2 refers to di- and tri-antennary N-glycans with two sialic acids, S3 refers to triantennary N-glycans with three sialic acids, Mannose refers to M4-M10 and hybrid-type N-glycans, Bisecting refers to any N-glycan with a bisecting GlcNAc moiety, Fucosylated refers to any N-glycan with a core-fucose. Healthy Control ($n = 2$), Day 4 Trajectory 1&2 ($n = 6$), Day 7 Trajectory 1&2 ($n = 5$), Day 4 Trajectory 3 ($n = 6$), Day 7 Trajectory 3 ($n = 5$), Day 4 Trajectory 4&5 ($n = 10$), Day 7 Trajectory 4&5 ($n = 6$). N-glycan classes are graphed as mean values +/– S.D. Statistical significance was determined using a one-way ANOVA with Tukey's multiple comparisons test *$p < 0.05$, ***$p < 0.001$. Source data are provided as a Source Data file.

correlated with MAN1A2, the o-mannosyltransferase TMTC2, and the α-2,3 sialyltransferase ST3GAL4 (Fig. 3b).

The expression of the α-2,3 sialyltransferase ST3GAL4 and the O-glycan α-2,6 sialyltransferase ST6GALNAC2 were significantly elevated in the severe COVID-19 cohort (Fig. 3a). Interestingly, the ST6GAL1 did not significantly differ between COVID-19 severity suggesting that a portion of the increased sialylation on IgM is due to the α-2,3 sialyltransferase ST3GAL4 (Supplemental Table 7). When IgM N-glycans were digested with the exoglycosidase neuraminidase S, specifically cleaving α-2,3-linked sialic acids, we detect a significant reduction in the A3G3S3 glycan species and an increase in the A3G3S2 abundance (Supplemental Fig. 5). Because ST6GALNAC2 adds an α-2,3 linked sialic acid to the O-glycans expressed on leukocyte cell surfaces, it is unlikely to add sialic acid to IgM[64]. However, the increased ST6GALNAC2 expression in the severe COVID-19 cohort PBMCs may reflect a reduced propensity for leukocytes to migrate into tissues due to sialic acid blocking P-/L-selectin ligand affinity[65]. Lastly, we report that a summation of all the sialic acids (S1, S2, and S3) from IgM positively correlated with the expression of ST3GAL4 (Fig. 3b). This finding suggests a potential role for ST3GAL4 adding sialic acid to IgM, but future studies will need to confirm this phenomenon specifically in plasmablast transcriptomic studies. In comparison to IgM N-glycans, IgG N-glycan agalactosylation, di-galactosylation, and mono-sialylation correlated to MAN1A2, but not ST3GAL4 or ST6GAL1 expression (Supplemental Tables 5, 6). All in all, the PBMC transcriptomic data support our observations of IgM glycosylation within the severe COVID-19 cohort.

## Clinical markers of COVID-19 severity correlate with IgM glycosylation

Next, we sought to determine if the changes in IgM N-glycosylation were associated with clinical laboratory data and additional cytokine panels collected in the Drexel University cohort with the IMPACC study[10,12]. These data were analyzed for correlations to IgM total mannose and S2 content using a linear regression model (Supplemental Tables 2 and 3). The reduction of IgM mannose in severe COVID-19 patients negatively correlated with increased D-dimer, blood urea nitrogen (BUN), creatinine, and potassium (K+) (Fig. 4a). In addition, the increased IgM S2 content positively correlated with the same clinical measurements—except for a nonsignificant correlation with potassium, $p = 0.186$ (Fig. 4b). In contrast, levels of agalactosylation, di-galactosylation, and mono-sialylation from total IgG correlated with the clinical laboratory values of D-dimer, but not BUN, creatinine, or potassium (Supplemental Tables 4–6).

The severity of COVID-19 has also been associated with higher anti-SARS-CoV-2 IgG and IgA antibody abundance at the time of hospital admission[66]. Therefore, we sought to correlate IgM glycosylation with the relative abundance of anti-SARS-CoV-2 nucleocasid (anti-N) immunoglobulins (see Supplemental Table 7 for all comparisons of anti-N immunoglobulin abundances between severe and nonsevere cohorts). Anti-N IgA relative abundance negatively correlated with IgM

mannose content, while the increase in IgM S2 content positively correlated with anti-N titers of IgA, IgM, and IgG relative abundance (Fig. 4c). This correlation could suggest IgM glycosylation reflects a specific plasmablast phenotype during severe COVID-19 that differs from the nonsevere cohort. On the other hand, levels of IgG agalactosylation, di-galactosylation, and mono-sialylation did not correlate with anti-N antibody abundance (Supplemental Tables 4–6).

Last, we examined data from a 32-plex cytokine panel to determine if circulating cytokines were associated with the glycosylation changes observed on IgM. Cytokines previously demonstrated to alter glycosyltransferase activity such as IFN-γ, TNF-α, IL-6, IL-17A, or IL-10[67,68] did not significantly correlate with IgM mannose or S2 content (Supplemental Tables 2 and 3). The only cytokine to increase significantly in the severe cohort compared to the nonsevere cohort was IL-18 (Supplemental Table 7). This cytokine nearly correlated ($p = 0.057$) to the IgM mannosylation levels (Supplemental Table 3). Taken together, IgM N-glycosylation correlates to other known markers of COVID-19 severity, but the factors inducing the changes to IgM N-glycosylation during severe COVID-19 have yet to be elucidated.

## SARS-CoV-2 spike S1-specific immunoglobulin N-glycan profiles

To further characterize the differences observed between the severe and nonsevere COVID-19 cohort, we analyzed N-glycan profiles from SARS-CoV-2 spike S1-specific immunoglobulins. These N-glycan profiles represent pooled plasma from 12 nonsevere and 10 severe COVID-19 patients. The heavy chain of spike S1-specific IgG from the severe cohort contained more FA1G0 (4.9-fold, GP1), FA2G0 (2-fold, GP3), and FA2G1 (1.7-fold, GP7) compared to the nonsevere cohort (Fig. 5a, Supplemental Table 8). Meanwhile, the A2G2S2 (GP21) N-glycan was over 2-fold more abundant in the nonsevere cohort spike S1-specific IgG heavy chain. These findings align with the severe versus nonsevere COVID-19 total IgG N-glycan profile analysis and previous reports of spike S1-specific IgG N-glycosylation[26].

We next analyzed the N-glycans associated with the spike S1-specific heavy chain of IgM. Here, spike S1-specific IgM from the severe cohort contained more FA2BG1 (1.6-fold, GP9), FA2G2 (1.6-fold, GP13), FA2G2S1 (1.4-fold, GP18), FA2G2S2 (1.2-fold, GP24), and total S3 (1.3-fold, GP30-33) N-glycan species compared to the nonsevere cohort (Fig. 5b, Supplemental Table 9). In comparison to the total IgM N-glycan profile, there was higher relative abundance in the A2G2S2 (GP23) species in both the severe and nonsevere cohorts.

Last, we sialidase-digested the severe and nonsevere spike S1-specific IgM N-glycans Because sialylated complex-type N-glycans elute at similar retention times as the high-mannose species M7-M10. By removing the sialylated species from the IgM N-glycan profile, we observed an increased abundance of high-mannose species M7 (1.1-fold, GP13-14), M8 (1.5-fold, GP16-19), M9 (1.3-fold, GP20-21), and M10 (2.1-fold, GP22) in the severe cohort (Fig. 5c, Supplemental Table 10). The increased abundance of M7-M10 N-glycans in the severe COVID-19 cohort concurs with the total IgM N-glycan profiles and the Golgi mannosidase expression data from the severe COVID-19 cohort. As

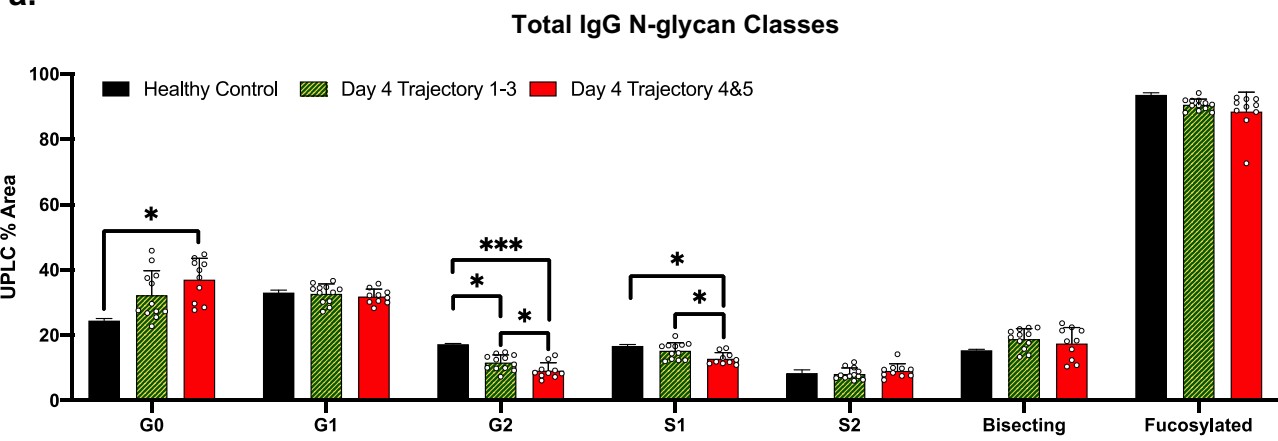

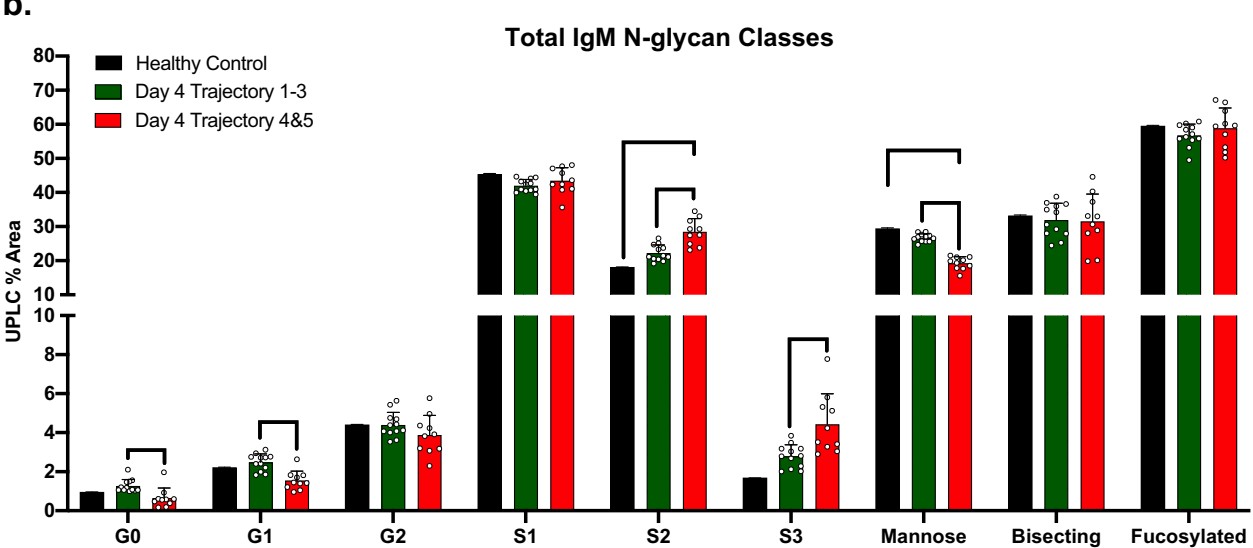

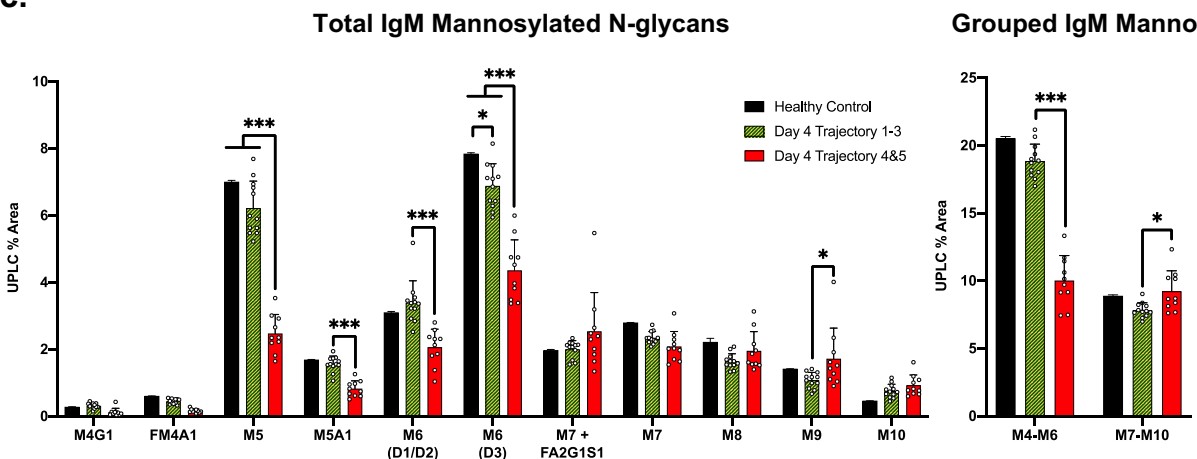

**Fig. 2 | IgM N-glycan profile stratifies cohorts of nonsevere from severe trajectory COVID-19 patients. a** IgG N-glycans from healthy control ($n = 2$), day 4 trajectory 1–3 ($n = 12$), and day 4 trajectory 4&5 ($n = 10$) cohorts. N-glycans are graphed as grouped classes--see supplemental Fig. 4 for a full list of N-glycans and N-glycan grouping. **b** IgM N-glycan profiles from cohorts of healthy control ($n = 2$), day 4 trajectory 1–3 ($n = 12$), and day 4 trajectory 4&5 ($n = 10$) hospitalized COVID patients. Data are presented as mean values +/− S.D. See Fig. 1c for a detailed explanation of N-glycan classes. **c** IgM mannosylated N-glycans from non-severe compared to severe COVID-19. A summation of the indicated mannose/hybrid N-glycan sub-groups are graphed to the right. IgM N-glycan classes are graphed as mean +/− S.D. Statistical significance was determined using two-sided unpaired t-tests *$p < 0.05$, **$p < 0.01$, ***$p < 0.001$. Source data are provided as a Source Data file.

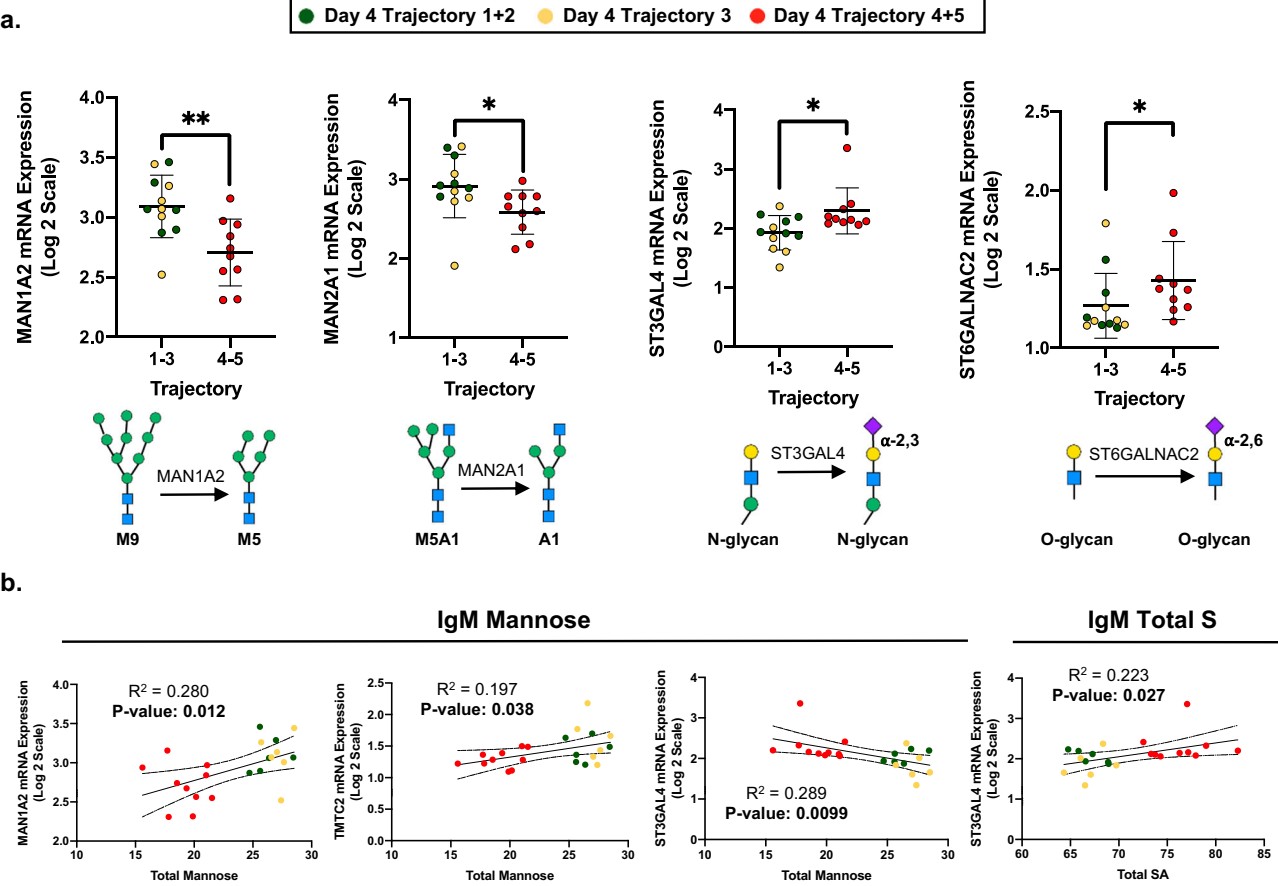

**Fig. 3 | Changes in IgM N-glycosylation correlate with PBMC glycosyltransferase/glycosidase mRNA expression. a** COVID-19 trajectory 1–3 (nonsevere, $n = 12$ biologically independent samples) and trajectory 4 and 5 (severe, $n = 10$ biologically independent samples) expression of glycosyltransferases were significantly different between MAN1A2 ($p = 0.007$), MAN2A1 ($p = 0.025$), ST3GAL4 ($p = 0.018$), and ST6GALNAC2 ($p = 0.025$). Data are presented as mean values +/− S.D. The role of each glycosidase and glycosyltransferase are depicted below. **b** Total mannose on IgM positively correlated with MAN1A2 and TMTC3 expression while negatively correlating with ST3GAL4 expression. The summation of sialic acids on IgM positively correlated with ST3GAL4 expression. mRNA expression is graphed as mean +/− S.D. Statistical significance was determined using a two-sided Kruskal-Wallis test with *$p < 0.05$ and **$p < 0.01$. Associations between IgM N-glycosylation and mRNA expression were determined using simple linear regression analysis. Source data are provided as a Source Data file.

indicated in the illustration of IgM heavy chains in Fig. 5b, c, these high-mannose structures likely occupy the Asn-563 and Asn-402 close to the C-terminus of the IgM constant domain and could participate in complement deposition. Taken together, the N-glycan trends from total plasma IgG and IgM are confirmed within the respective spike S1-specific immunoglobin G and M N-glycan profiles and offer new insights into the acute viral response.

### Increased IgM-dependent ADCD in severe COVID-19 patients

We next sought to interrogate the differences in complement deposition rates initiated by SARS-CoV-2 circulating plasma antibodies in general, and IgM specifically. We adapted an antibody-dependent complement deposition (ADCD) assay employing fluorescent beads conjugated to a biotinylated antigen to compare complement deposition rates with SARS-CoV-2 antigens: receptor binding domain (RBD), and spike S1[69]. After incubating either diluted plasma or purified IgM with antigen-coated beads, deposition of guinea pig complement was detected using flow cytometry (Fig. 6a). In nonsevere and severe plasma, RBD induced low ADCD, aligning with previously reported ADCD trends[70,71] (Fig. 6b). Further, purified IgM incubated with RBD did not induce complement deposition above the PBS background control.

We next assayed ADCD with the spike S1 antigen because others had detected higher levels of complement deposition by using the whole length of the spike S1 rather than the RBD[71]. Plasma from severe and nonsevere COVID-19 cohorts incubated with spike S1 deposited levels of complement above background, with the severe cohort depositing slightly higher levels (Fig. 6c). We postulate that the IgG in these plasma samples is the major determinant of complement deposition due to its antigen affinity, and the higher relative abundance of IgG in plasma[72]. Nevertheless, when equal amounts of purified total IgM were incubated with spike S1 from the severe and nonsevere COVID-19 cohorts, we observed a 2.8-fold higher rate of IgM-dependent complement deposition in the severe COVID-19 cohort (68.5%) versus the nonsevere cohort (24.3%) (Fig. 6c). Of note, the abundance of anti-spike S1 IgM from the pooled severe COVID-19 cohort was 1.7-fold higher than the nonsevere cohort as indicated by LC-MS/MS analysis of the Coomassie-stained Sodium dodecyl-sulfate polyacrylamide gel electrophoresis (SDS-PAGE) of spike S1-specific immunoglobulins (Supplemental Figs. 6c and d). This suggests the higher relative abundance of anti-spike S1 IgM and the N-glycans populating IgM responding to severe SARS-CoV-2 infection both promote higher complement deposition rates.

To explore the role N-glycans play during complement deposition, plasma and purified IgM were digested with a non-specific sialidase (S) and assayed for spike S1-specific ADCD (Fig. 6d). Plasma spike S1 ADCD remained unaffected following sialidase digestion; suggesting the predominant immunoglobulin in plasma, IgG, does not require sialic acid to promote complement deposition. In contrast, sialidase digestion of spike S1-specific IgM from the severe COVID-19 cohort

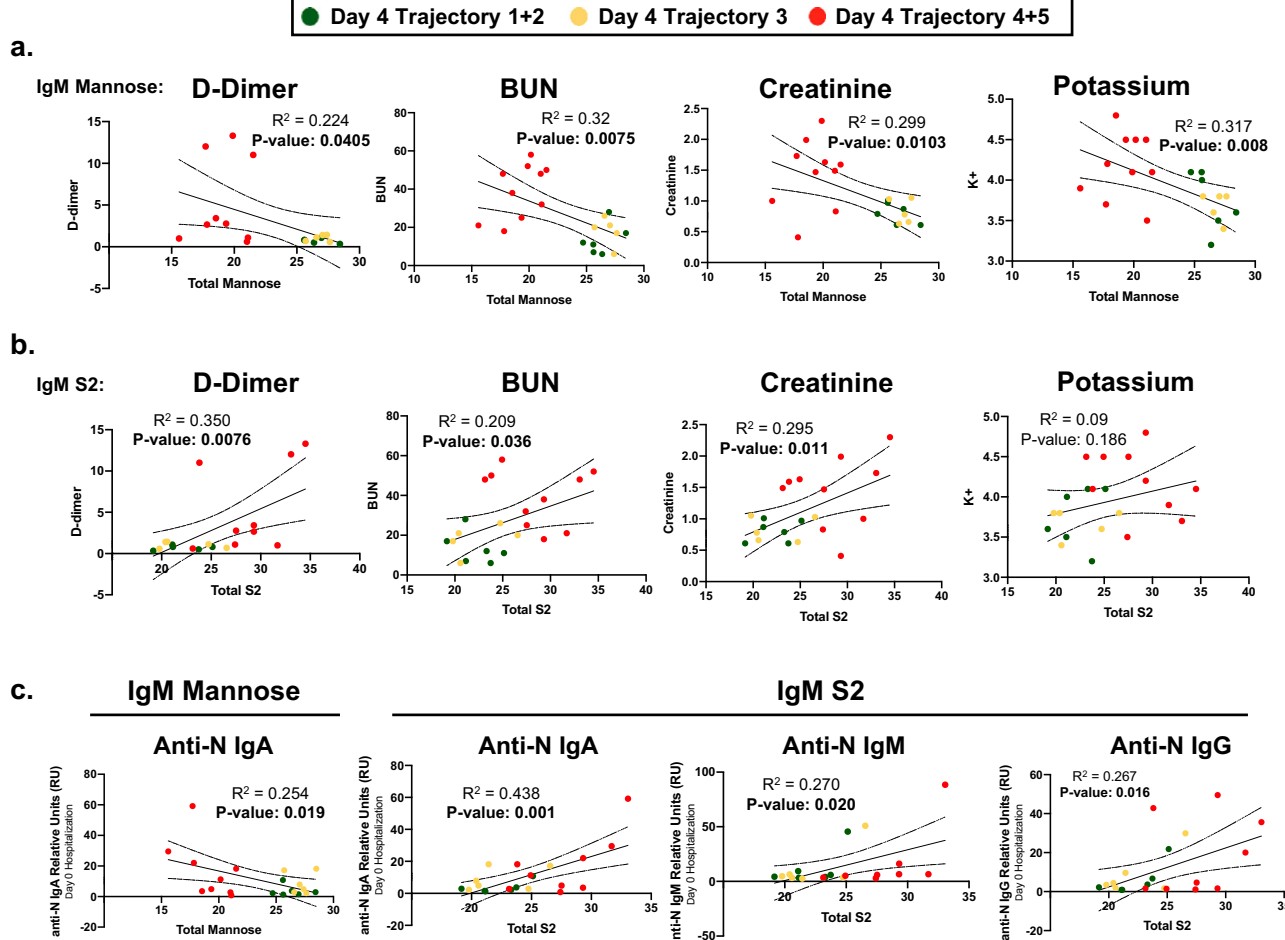

**Fig. 4 | Changes in IgM N-glycosylation Associate with Clinical Markers of COVID-19 Severity. a** Total mannose content (summation of M4-M10 and hybrid N-glycans) was correlated to hospital laboratory measurements of D-dimer, Blood urea nitrogen (BUN), creatinine, and potassium measured on day 4 of hospitalization using simple linear regression analysis. $R^2$ and p-values are reported for each comparison, estimated by simple linear regression, with bolded p-values considered statistically significant. **b** Total di-sialylated (S2) N-glycans were correlated with hospital laboratory measurements of D-dimer, BUN, creatinine, and potassium using simple linear regression. $R^2$ and *p*-values are reported for each comparison, estimated by simple linear regression, with bolded *p*-values considered statistically significant. **c** Anti-nucleocapsid protein (anti-N) IgA, IgM, and IgG detected from patient plasma donated at the time of hospital admission (Day 0) were correlated to IgM mannose content and S2 content. Green dots identify day 4 Trajectory 1 + 2, yellow dots identify day 4 trajectory 3, and red dots identify day 4 trajectory 4 + 5 hospitalized COVID-19 cohorts. $R^2$ and *p*-values are reported for each comparison, estimated by simple linear regression, with bolded *p*-values considered statistically significant. Source data are provided as a Source Data file.

reduced complement deposition by 50% compared to the undigested severe cohort IgM. The remaining 34.7% complement deposition in the sialidase-digested severe IgM could be related to the increased M7-M10 content associated with severe COVID-19. Taken together, we report that severe COVID-19 cohort IgM induces higher levels of antigen-specific complement deposition – which could be glycosylation-dependent.

## Discussion

IgG N-glycosylation and effector function have been well characterized during acute SARS-CoV-2 infection[26–30]. However, IgM antibodies also play vital roles during immune responses, promote affinity maturation, maintain hemostasis at mucosal sites including the gut and lung, and induce significantly higher levels of complement deposition compared to IgG[73]. We suggest that IgM N-glycosylation has been overlooked during the acute COVID-19 immune response. Within the subset of the IMPACC cohort enrolled at Drexel University (*n* = 22), we find that host IgM N-glycosylation correlates with disease severity and may promote antigen-specific complement deposition.

We report a significant decrease in total IgM mannose in patients with severe COVID-19 (trajectories 4 and 5) compared to those with

nonsevere COVID-19 (trajectories 1-3). By examining the mannose and hybrid structures contributing to this decrease, we conclude IgM contains fewer (M4-M6) mannose structures during severe COVID-19. Instead, IgM in severe COVID-19 contains larger mannose structures. The observation of increased levels of high-mannose M7-M10 were confirmed when analyzing Spike S1-specific IgM. These findings are supported by decreased mannosidase MAN2A1 and MAN1A2 expression within patient PBMC mRNA glycosyltransferase (GT) expression responsible for trimming high-mannose content in the Golgi. Previously, MAN1A2 genetic variability was identified as a potential correlate with susceptibility to SARS-CoV-2 infection[74] and is regulated by miRNA during influenza[75,76]. More work into the regulation of mannosidase expression is required to confirm if the changes observed in PBMC mRNA are maintained within the plasmablast cell population.

Considering the increased sialic acid content on total and spike S1-specific IgM from patients with severe COVID-19, one publication reported higher levels of sialic acid detected in IgM isolated from cancer patients[49]. During severe COVID-19, increased sialylation is likely presented on IgM glycosylation sites: Asn-395, Asn-332, and Asn-171 which could participate in immunomodulatory signaling. Multiple receptors on immune cells may interact with IgM presenting increased

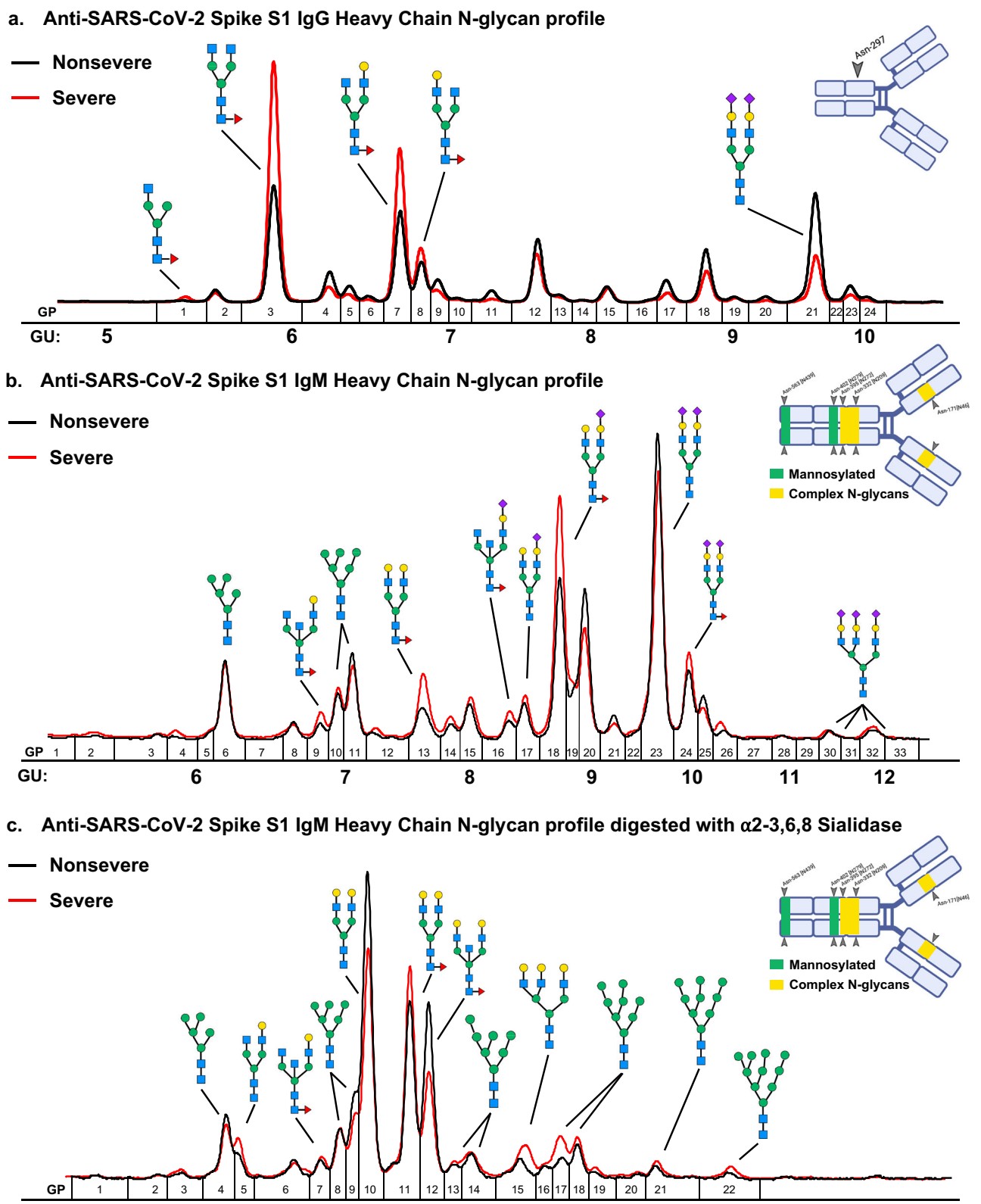

**Fig. 5 | Assessment of Day 4 SARS-CoV-2 spike S1-specific immunoglobulin G and M heavy chain N-glycosylation. a** Fluorescent N-glycan traces are overlaid from day 4 pooled severe (red line) and day 4 nonsevere (black line) anti-SARS-CoV-2 spike S1 IgG heavy chain (50 kDa) isolated from SDS-PAGE gel plugs **b** Fluorescent N-glycan traces are overlaid from day 4 pooled severe (red line) and day 4 nonsevere (black line) anti-SARS-CoV-2 spike S1 IgM heavy chain (75 kDa) isolated from SDS-PAGE gel plugs. **c** Sialidase digested fluorescent N-glycan traces are overlaid from day 4 pooled severe (red line) and day 4 nonsevere (black line) anti-SARS-CoV-2 spike S1 IgM heavy chain (75 kDa) isolated from SDS-PAGE gel plugs. Glycan peak (GP) number and Glucose units (GU) are indicated under each immunoglobulin heavy chain trace. N-glycosylation sites of IgG and IgM are presented in the upper right corner of each panel, created using BioRender. Green bars indicate high-mannose content while yellow bars indicate complex-type glycosylation sites on the IgM heavy chain. N-glycans with visible differences between severe and non-severe cohort N-glycan profiles are displayed.

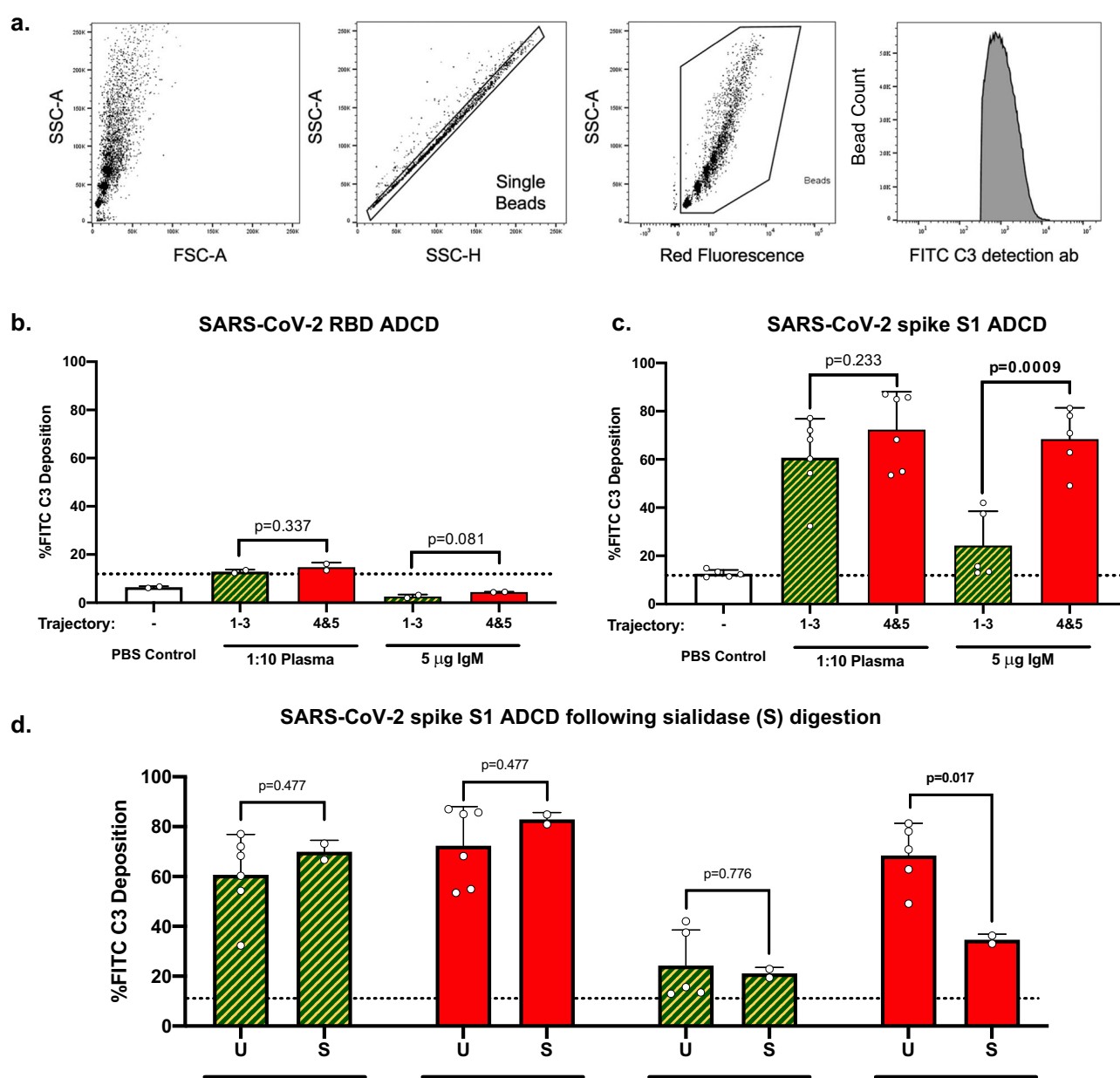

**Fig. 6 | Antigen-specific complement deposition (ADCD) induced by plasma and IgM from severe and nonsevere COVID-19 cohorts. a** Gating strategy for the detection of complement deposition on fluorescent beads using flow cytometry. **b** ADCD assay using the RBD antigen assayed two times with pooled day 4 trajectory 1–3 ($n = 1$ biologically independent sample) and pooled day 4 trajectory 4&5 ($n = 1$ biologically independent sample) plasma or IgM. Data are presented as mean values +/− S.D. **c** SARS-CoV-2 spike S1 antigen was assayed for ADCD with pooled day 4 trajectory 1–3 ($n = 1$ biologically independent sample) and pooled day 4 trajectory 4&5 ($n = 1$ biologically independent sample) plasma six times. Spike S1 antigen was assayed for ADCD with pooled day 4 trajectory 1-3 ($n = 1$ biologically independent sample) and pooled day 4 trajectory 4&5 ($n = 1$ biologically independent sample) IgM five times. Data are presented as mean values +/− S.D. **d** Plasma and IgM samples from pooled day 4 trajectory 1–3 ($n = 1$ biologically independent sample) and pooled day 4 trajectory 4&5 ($n = 1$ biologically independent sample) remained undigested (U) before assaying for ADCD six times for plasma and five times for IgM. Samples of plasma and IgM from day 4 trajectory 1–3 ($n = 1$ biologically independent sample) and pooled day 4 trajectory 4&5 ($n = 1$ biologically independent sample) were digested with sialidase (S) before assaying for ADCD two times. Data are presented as mean values +/− S.D. Dotted horizontal lines refer to background binding by FITC anti-C3 antibody in PBS-only samples. Statistical significance was determined using a two-sided unpaired t-test. Source data are provided as a Source Data file.

sialic acid content, resulting in functional consequences for the humoral immune response. For example, sialic acid-binding Ig-type lectin G (Siglec G or CD22) expressed on B-cells has been reported to bind IgM[77].

When we examined the PBMC sialyltransferase mRNA expression data, we did not observe significant changes in the ST6GAL1 mRNA levels between severe and nonsevere COVID-19 cohorts. However, we detected increased ST3GAL4 mRNA expression in the severe COVID-19 cohort, which positively correlated with the summation of all sialic acid content on IgM. We observed the presence of α-2,3 sialylation on IgM, suggesting some of the increased sialylation on IgM isolated from severe COVID-19 patients could be attributed to ST3GAL4 activity. A

previous high-throughput glycomic analysis of COVID-19 patients identified increased α-2,6 and α-2,3 sialylation in total plasma, lung, and liver tissue[78]; associating the increased sialylation of complement proteins with heightened rates of complement deposition during severe COVID-19. Therefore, the increase in ST3GAL4 during severe COVID-19 may be exerting proinflammatory effects during severe COVID-19 pathogenesis through IgM and other glycoproteins.

In severe and nonsevere COVID-19 cohorts, we observed significant increases in the agalactosylated N-glycans on IgG with concomitant decreases of G2 and S1. These findings were confirmed in the spike S1-specific IgG N-glycan profile, with higher levels of agalactosylation in the severe COVID-19 cohort. In contrast, the IgM N-glycan profile decreased G0 and G1 content, instead increasing S2 and S3 sialic acid as well as acquiring larger, unprocessed mannose content. These trends were observed in the spike S1-specific IgM subpopulation. The differences in glycosylation detected between IgG and IgM from the same COVID-19 patients suggest that IgG and IgM are processed in the Golgi in different manners during severe COVID-19. Because IgG contains nearly all α-2,6 sialic acid[79] and ST6GAL1 transcripts remain unchanged between severe and nonsevere COVID-19 cohorts, the upregulation of ST3GAL4 may contribute to the increased sialylation observed on IgM in severe COVID-19 patients. Glycosyltransferase expression is regulated by multiple cytokine and chemokine factors during an immune response, but these regulatory factors are not fully elucidated[80]. Taken together, the differences in total and antigen-specific immunoglobulin G versus M glycosylation require further investigation and may reveal more details about the humoral response to severe viral infections.

To understand the factors promoting COVID-19 severity, markers of severe COVID-19 were correlated with IgM N-glycans from severe and nonsevere COVID-19 cohorts. Elevated D-dimer, blood urea nitrogen (BUN), creatinine, potassium, and the abundance of anti-nucleocapsid antibodies[81] correlated with mannose and S2 IgM N-glycosylation. Elevations in potassium, BUN, and creatinine reflect acute kidney injury[82–84] while elevated D-dimer indirectly reflects circulatory thrombosis often observed in severe COVID-19 patients[4,85]. In contrast, IgG N-glycans from severe and nonsevere cohorts correlated with the elevation in D-dimer but no other clinical parameters. In the future, it would be fascinating to examine IgM and IgG N-glycosylation in patients with long-COVID in the post-acute setting, building off work indicating total IgM and IgG3 signatures predict post-acute COVID-19 syndrome[86]. All in all, IgM N-glycans correlated with multiple markers of COVID-19 severity and may play a role in severe COVID-19 pathogenesis.

Overactivation of complement has been associated with mortality and morbidity from COVID-19 in severe cases[70,87–90]. Of note, a recent multi-omic analysis of over 500 COVID-19 patients from the IMPACC study identified complement activation as a contributor to the maintenance of a severe inflammatory response to SARS-CoV-2[11]. Because IgM is highly effective at inducing complement, and the N-glycans on IgM were significantly altered in severe vs nonsevere COVID-19 trajectories, we sought to determine if the N-glycans on IgM impact SARS-CoV-2 antigen-specific complement deposition. The RBD antigen complement deposition was low, likely due to the lower levels of specific anti-RBD antibody abundance during the first 10 days in COVID-19 naïve patients lacking previous vaccinations[11]. When we assayed ADCD with the spike S1 antigen, we observed higher complement deposition in the plasma of the severe COVID-19 cohort compared to the nonsevere cohort. We next observed that purified IgM from the severe COVID-19 cohort led to significant complement deposition when incubated with the spike S1 antigen. Of note, purified IgM has been previously assayed for complement deposition using guinea pig complement[91,92]. Compared to IgM from the nonsevere COVID-19 cohort, the severe cohort IgM induced significantly higher levels of spike S1 ADCD. Because IgM interacts with complement

C1q[42,93], we hypothesize that the sialic acid and high-mannose N-glycans associated with the severe cohort spike S1-specific IgM impact the rate of antigen-specific complement deposition.

We digested plasma and IgM with a non-specific sialidase to explore the role sialic acids play in ADCD. Spike S1 ADCD from plasma digested with a sialidase remained high, indicating the predominant plasma immunoglobulin, IgG, does not require sialic acid to induce complement deposition[94]. However, spike S1 ADCD was reduced when IgM from severe COVID-19 was digested with a non-specific sialidase. This suggests sialylated IgM N-glycans are required for optimal complement deposition, and the remaining complement deposition could be associated with the high-mannose content detected on IgM from the severe COVID-19 cohort. It is intriguing to see that IgM glycosylation could be in part responsible for promoting complement deposition during severe COVID-19 pathogenesis. We hypothesize that complement deposition by IgM in conjunction with IgG, could promote acute respiratory distress syndrome (ARDS) or acute kidney injury (AKI) observed in severe COVID-19 patients. In the future, larger sets of individual severe and nonsevere COVID-19 patient cohorts should be assayed for ADCD to confirm these findings.

This report analyzed patients from Drexel University's portion of the IMPACC study. Larger sample sets collected from multiple hospital sites should confirm these findings. Furthermore, this cohort was collected early in 2020 when COVID-19 was predominantly driven by the ancestral SARS-CoV-2 strain. Patients at this time lacked access to life-saving vaccines, antiviral medications, and rapid testing. Therefore, newer variants of the virus, more effective treatments, and vaccination may alter the characteristics of severe COVID-19 patient IgM N-glycosylation. In addition, one extraneous source of N-glycans is the IgM pentamer J-chain. Yet, only one out of the ~60 N-glycans per IgM pentamer is associated with the J-chain and thus this potential N-glycan contribution was ignored during data analysis. Lastly, antigen-specific IgM glycopeptide analysis would provide more information about the site-specific glycosylation response to severe infections.

In conclusion, IgM N-glycosylation changes in interesting and unexpected ways compared to IgG N-glycans in severe COVID-19 patients. The identification, quantification, and correlation of the IgM N-glycan profile within a well-characterized cohort provided opportunities to learn more about how the human immune system responds to an acute viral infection. We align glycosyltransferase expression to the increased mannose complexity and sialic acid content on IgM and contrast these findings to what is canonically observed in IgG N-glycan profiles from patients with severe COVID-19. Spike S1-specific IgG and IgM N-glycan profiles confirmed our observations from total immunoglobulin N-glycan analysis. We correlate the IgM N-glycan profile to markers of disease severity and report that spike S1 specific complement deposition driven by IgM may contribute to severe COVID-19 pathophysiology. A better understanding of IgM N-glycosylation could one day result in novel therapeutics to reduce the severity of acute infectious diseases in humans. Taken together, this data opens the field for immunoglobulin M N-glycans to be characterized during other infectious disease states.

## Methods
### Human samples
**Ethics**. NIAID staff conferred with the Department of Health and Human Services Office for Human Research Protections (OHRP) regarding the potential applicability of the public health surveillance exception [45CFR46.102(l) (2)] to the IMPACC study protocol. OHRP concurred that the study satisfied criteria for the public health surveillance exception, and the IMPACC study team sent the study protocol, and participant information sheet for review, and assessment to institutional review boards (IRBs) at participating institutions. Twelve institutions elected to conduct the study as public health surveillance, while 3 sites with prior IRB-approved biobanking protocols elected to

integrate and conduct IMPACC under their institutional protocols (University of Texas at Austin, IRB 2020-04-0117; University of California San Francisco, IRB 20-30497; Case Western Reserve University, IRB STUDY20200573) with informed consent requirements. Participants enrolled under the public health surveillance exclusion were provided information sheets describing the study, samples to be collected, and plans for data de-identification, and use. Those that requested not to participate after reviewing the information sheet were not enrolled. In addition, participants did not receive compensation for study participation while inpatient, and subsequently were offered compensation during outpatient follow-ups.

**Patient enrollment and consent.** IMPACC is a collaborative project developed by NIAID and investigators from the Human Immunology Project Consortium (HIPC), the Asthma and Allergic Diseases, and the Cooperative Disease Research Centers (AADCRC) and other NIAID-funded investigators. Drexel University collected patient samples to be included in the IMPACC through the Tower Health Hospital network from May 2020 to March 2021. During this enrollment period, COVID-19 vaccines were not widely available. Participants were enrolled within 72 hours of hospitalization under the public health surveillance exception. Upon enrollment, demographics, COVID-19 symptoms, detailed medical history (including comorbidities), clinical laboratory data, and imaging data were collected. Patients were confirmed with a positive SARS-CoV-2 polymerase chain reaction (PCR). Biological samples including blood, nasal swab, and endotracheal aspirates (when available) were collected. Clinical data and samples from days 4 and 7, representing patient admission to the hospital, were examined[10]. Table 1 provides information on patient sex, age, and demographics sorted by trajectory collected from medical records. This cohort was not powered to perform sex or gender analysis.

**Biological sample processing.** Blood samples and nasal swabs were collected at each timepoint and processed at Drexel University within 6 hours of collection according to the IMPACC standardized operating procedure and analyzed at Drexel under the IRB protocols 2004007753 and 2102008337[10]. Whole blood, nasal swabs, peripheral blood mononuclear cells (PBMCs), and plasma collected from each patient were processed at Drexel University and sent to IMPACC core facility sites for further analysis as previously reported[10,12]. PBMCs were used to identify immune cell populations and changes in cell populations, gene expression, and activation markers. Plasma was used to characterize antibody titers, anti-RBD titers, antibody isotype, proteomics, and metabolomics. At Drexel, plasma was additionally used for enzyme-linked immunosorbent assay (ELISA) antibody abundance analysis, Luminex cytokine and chemokine assays, and glycomic analysis. Whole blood was used in a genome-wide association study (GWAS) and cytometry by time-of-flight (CyTOF) and bulk RNA transcriptomics. Nasal Swabs were used for bulk RNAseq and viral load quantitation. Patient plasma used in this study were selected prior to the start of analysis to ensure a balanced sex, age, and BMI across trajectories.

**PBMC isolation.** Patient blood samples were spun down at 1000 x g for 10 minutes at room temperature, and plasma was aliquoted. The remaining blood was diluted 1:2 with Dulbecco's phosphate-buffered saline (DPBS, Ca$^{+2}$Mg$^{+2}$ free) and slowly pipetted into a 50 mL SepMate-50 tube (with 15 mL Lymphoprep below the insert). Samples were spun at 800 x g for 20 minutes at 20 °C with brakes off. The top layer with PBMCs was transferred to a new tube and cells were washed at 400 x g for 5 minutes. Cells were resuspended in 20 mL EasySep Buffer, then spun again at 300 x g for 10 minutes at room temperature. For RNA-Seq, cells were resuspended at 5 million per mL, and 50uL was aliquoted into CRYSTAL Gen tubes. Cells were spun at 500 x g for 5 minutes at room temperature and the excess media was removed.

200 uL QIAGEN RLT Buffer with (BME) was added and cells were vortexed until the pellet was fully dissolved. Samples were stored at −80 °C for shipment. The remaining PBMCs were frozen down in fetal bovine serum (FBS) and dimethyl sulfoxide (DMSO) for storage at Drexel University.

**Anti-SARS-CoV-2 nucleocapsid IgA, IgG, and IgM quantitation**
Monobind AccuBind® ELISA Anti-SARS-CoV-2 kits were used as a qualitative determination of Anti-SARS-CoV-2 specific IgA, IgG and IgM antibodies at Drexel's IMPACC site. These kits utilize a sequential sandwich ELISA method. This test utilizes recombinant nucleocapsid protein (rNCP) from SARS-CoV-2 coated on microwells to capture antibodies in human plasma. Patient plasma was diluted 1:100 and added directly to the ELISA plate. Following incubation and washing, IgA, IgG or IgM labeled antibodies were added. After a second incubation and wash, reagent substrate was added to produce a measurable color through the reaction with enzyme and hydrogen peroxide. After the addition of a stop substrate, absorbance was read in each well at 450 nm within 15 minutes of adding the stop solution.

**Cytokine and chemokine analysis**
Patient plasma was analyzed for chemokine/cytokine levels using the human immune monitoring 65-Plex ProcartaPlex™ Panel (Invitrogen™). This kit was used to determine the levels of 65 cytokines, chemokines, growth factors, and soluble receptors produced at the designated time points at the Drexel IMPACC site. The following human chemokine/cytokine premixed panel was used according to the manufacturer's protocol: G-CSF (CSF-3), GM-CSF, IFN alpha, IFN-g, IL-1a, IL-1b, IL-2, IL-3, IL-4, IL-5, IL-6, IL-7, IL-8 (CXCL8), IL-9, IL-10, IL-12p70, IL-13, IL-15, IL-16, IL-17A (CTLA-8), IL-18, IL-20, IL-21, IL-22, IL-23, IL-27, IL-31, LIF, M-CSF, MIF, TNF-a, TNF-b, TSLP, BLC (CXCL13), ENA-78 (CXCL5), Eotaxin (CCL11), Eotaxin-2 (CCL24), Eotaxin-3 (CCL26), Fractalkine (CX3CL1), Gro-alpha (CXCL1), IP-10 (CXCL10), I-TAC (CXCL11), MCP-1 (CCL2), MCP-2 (CCL8), MCP-3 (CCL7), MDC (CCL22), MIG (CXCL9), MIP-1a (CCL3), MIP-1b (CCL4), MIP-3a (CCL20), SDF-1a (CXCL12), FGF-2, HGF, MMP-1, NGF-b, SCF, VEGF-A, APRIL, BAFF, CD30, CD40L (CD154), IL-2R (CD25), TNF-RII, TRAIL (CD253), TWEAK. Data was acquired on a Luminex™ FLEXMAP 3D™ System using bead regions defined in the protocol and analyzed using Belysa Curve Fitting Software (Sigma Aldrich). Standard curves were generated, and sample concentrations were calculated in pg/mL.

**Nasal viral PCR, host transcriptomics, and metagenomics**
Nasal viral PCR, host transcriptomics, and metagenomics were performed as detailed in *Diray-Arce* et al. 2023[11]. The ImmPort accession for the study data is SDY1760. Brief descriptions of each method are listed below.

**RNA preparation.** Inferior nasal turbinate swabs were collected and placed in 1 ml of Zymo-DNA/RNA shield reagent (Zymo Research). RNA was extracted from 250 µL of sample and eluted into a volume of 50ul using the KingFisher Flex sample purification system (ThermoFisher) and the quick DNA-RNA MagBead kit (Zymo Research) following the manufacturer's instructions. Each sample was extracted twice in parallel. The 2 eluted RNA samples were pooled and aliquoted into 20 µL aliquots using a Rainin Liquidator 96 pipettor for downstream RT-qPCR, RNA-sequencing, and viral sequencing.

**RealTime quantitative polymerase chain reaction.** Master mixes containing nuclease-free water, combined primer/probe mixes, and One-Step RT675 qPCR ToughMix (Quantabio) were prepared on ice, and 15 µL was dispensed in each well of a 384-reaction plate (Thermo Fisher) SARS-CoV-2 genome was quantitated using the CDC qRT-PCR assay[95] (primers and probes from IDT). Briefly, this comprised two reactions targeting the SARS-CoV-2 nucleocapsid gene (N1 and N2) and

one reaction targeting RPP30 (RP). Each batch included positive controls of plasmids containing N1/N2 and RP target sequence (2019-nCoV_N_Positive Control and Hs_RPP30 Positive Control, IDT) to allow quantitation of each transcript. Primer/probe sequences were: 2019-nCOV_N1-F GAC CCC AAA ATC AGC GAA AT, 2019-nCOV_N1-R TCT GGT TAC TGC CAG TTG AAT CTG, 2019-nCOV_N1-P ACC CCG CAT TAC GTT TGG TGG ACC, 2019-nCOV_N2-F TTA CAA ACA TTG GCC GCA AA, 2019-nCOV_N2-R GCG CGA CAT TCC GAA GAA, 2019-nCOV_N2-P ACA ATT TGC CCC CAG CGC TTC AG, RP-F AGA TTT GGA CCT GCG AGC G, RP-R GAG CGG CTG TCT CCA CAA GT and RP-P TTC TGA CCT GAA GGC TCT GCG CG. After RNA extracts were gently vortexed and added 5 µL per sample. Plates were centrifuged for 30 s at 500 x *g*, 4 C. The quantitative polymerase chain reaction was performed using a Quantstudio5 (Thermo Fisher) with cycling conditions: 1 cycle 10 min at 50 °C, followed by 3 min at 95 °C, 45 cycles 3 s at 95 °C, followed by 30 s at 55.0 °C.

**RNA-sequencing cDNA library production.** From each nasal RNA sample, 10ul was aliquoted to a library construction plate using the Perkin 692 Elmer Janus Workstation (Perkin Elmer, Janus II). Ribosomal depletion, cDNA synthesis, and library construction steps were performed using the Total Stranded RNA Prep with Ribo-Zero Plus kit, following the manufacturer's instructions (Illumina). All steps were automated on the Perkin Elmer Sciclone NGSx Workstation to reduce batch-to-batch variability and increase sample throughput. Final cDNA libraries were quantified using the Quant-it dsDNA High Sensitivity assay, and library insert size distribution was checked using a fragment analyzer (Advanced Analytical; kit ID DNF474). Samples, where adapter dimers constituted more than 4% of the electropherogram area, were failed before sequencing. Technical controls (K562, Thermo Fisher Scientific, cat# AM7832) were compared to expected results to ensure that batch-to-batch variability was minimized. Successful libraries were normalized to 10 nM for sequencing.

**RNA-sequencing clustering and sequencing.** Barcoded libraries were pooled using liquid handling robotics prior to loading. Massively parallel sequencing-by-synthesis with fluorescently labeled, reversibly terminating nucleotides was carried out on the NovaSeq 6000 sequencer using S4 flowcells with a target depth of 50 million 100 base-pair paired-end reads per sample (25 million read pairs).

## Total IgG isolation

Total IgG was isolated from 20 µL of plasma using a protein G spin plate as described by the manufacturer (ThermoFisher, MA). Four 200 µL 1X PBS washes removed unbound plasma protein using a vacuum manifold apparatus. Next, IgG was eluted by incubating 150 µL of 0.1 M glycine HCl pH 2–3 for 5 minutes at room temperature. The eluate was collected into a 96-well 2 mL collection plate pre-loaded with 15 µL of 1.5 M Tris pH 8 to neutralize the glycine elution buffer. The wash process was repeated a second time to ensure a high yield of IgG. The resulting 315 µL of the neutralized eluate was concentrated and buffer-exchanged to 20 µL of 1X PBS using Amicron Ultra-0.5 centrifugal Filter 10 kDa MWCO (Millipore) following the manufacturer's instructions. NanoDrop 1000 spectrophotometer readings monitored protein yield through the isolation process. Coomassie-stained SDS-PAGE gel and anti-IgG western blot confirmed IgG isolation during method development (Supplemental Fig. 6A) using Goat anti-human IgG IR680LT (LiCor, 926-68032, Lot #: D00421-13) following the protocol described previously[96].

## Total IgM isolation

Total IgM was isolated from plasma by incubating 80 µL of goat anti-IgM agarose-conjugated agarose beads (Sigma, A9935-5ML, Lot#: 0000188278) with 80 µL plasma and 100 µL 1X PBS for 2 hours at room temperature. Following the incubation, the solution was transferred to a 1.2 µm MultiScreen HTS 96-well filter plate. Four 200 µL 1X PBS washes removed unbound plasma protein using a vacuum manifold apparatus. Next, IgM was eluted by incubating 150 µL of 0.1 M glycine HCl pH 2-3 for 5 minutes at room temperature. The eluate was collected into a 96-well 2 mL collection plate pre-loaded with 15 µL of 1.5 M Tris pH 8 to neutralize the glycine elution buffer. The wash process was repeated a second time to ensure a high yield of IgM. The resulting 315 µL of the neutralized eluate was concentrated and buffer-exchanged to 20 µL of 1X PBS using Amicron Ultra-0.5 centrifugal Filter 10 kDa MWCO (Millipore) following the manufacturer's instructions. NanoDrop 1000 spectrophotometer readings monitored protein yield through the isolation process. Coomassie-stained SDS-PAGE gel and anti-IgM western blot confirmed IgM isolation during method development (Supplemental Fig. 6B), using 1:5,000 Anti-Human IgM (µ-chain specific) antibody produced in goat (Sigma, I2386-1ML, Lot #: 118M4782V) detected by 1:10,000 Donkey anti-Goat IgG IR800CW (LiCor, 926-32214, Lot# B70416-02) following the protocol described previously[96].

## SARS-CoV-2 specific immunoglobulin isolation

Streptavidin Agarose Resin (Thermo Scientific, 20347) was incubated 1:1 with SARS-CoV-2 (2019-nCoV) spike S1-His Recombinant Protein, Biotinylated (SinoBiological) for 1 hour at room temperature. The agarose resin was washed twice with 1X PBS and incubated with 1X CarboFree Block (Vector Laboratory) for 1 hour at room temperature followed by two 1X PBS washes. Next, 600 µL of pooled plasma from day 4 nonsevere and severe cohorts were diluted 1:5 in 1X PBS and incubated with aliquots of the spike S1-conjugated resin in a Multi-Screen HTS 96-well 0.2 µm filter plate for 3 hours at room temperature. Six 200 µL 1X PBS washes removed unbound plasma proteins using a vacuum manifold apparatus. Next, spike S1-specific immunoglobulins were eluted by incubating 150 µL of 0.1 M glycine HCl pH 2–3 for 5 min at room temperature. The eluate was collected into a 96-well 2 mL collection plate pre-loaded with 15 µL of 1.5 M Tris pH 8 to neutralize the glycine elution buffer. The wash process was repeated a second time to ensure a high yield of spike S1-specific immunoglobulins. The resulting neutralized eluate was concentrated and buffer-exchanged to 20 µL of 1X PBS using Amicron Ultra-0.5 centrifugal Filter 10 kDa MWCO (Millipore) following the manufacturer's instructions. NanoDrop 1000 spectrophotometer readings monitored protein yield through the isolation process. SARS-CoV-2 spike S1-specific immunoglobulin classes were isolated using 1D gel electrophoresis, visualized with Coomassie stain, and the 75 kDa (IgM) and 50 kDa (IgG) heavy chain bands were excised. Following de-staining, the glycans from each gel band were enzymatically removed and fluorescently labeled following standard in-gel PNGase F and labeling protocols as previously described[97]. The predominant proteins in the spike S1-specific 75 kDa and 50 kDa gel plugs were confirmed by trypsin digestion followed by LC-MS/MS peptide identification (Supplemental Figs. 6C and 6D).

## Immunoglobulin N-glycan analysis

N-glycans from IgG and IgM were released, labeled, and analyzed as described previously using the Waters GlycoWorks RapiFluor MS kit, adapted for PCR tubes[98]. Briefly, samples were denatured using the RapiGest reagent for 5 minutes at 95 °C using a PCR thermocycler. Next, glycoprotein samples were deglycosylated using PNGase F for 6 minutes at 60 °C using a PCR thermocycler. Afterward, samples were labeled with the RapiFluor mass spectrometry label (RFMS) for 5 minutes at room temperature. A solid-phase extraction (SPE) clean-up module isolated RFMS labeled N-glycans which were then eluted into a 96-well 2 mL Waters American National Standards Institute (ANSI) plate capped with a Polytetrafluoroethylene (PFTE) 96-well membrane top for high-throughput N-glycan analysis. An ACQUITY Premier Ultra Performance Liquid Chromatography (UPLC) System

was used following the setting and protocol described previously[98]. Briefly, an ACQUITY UPLC ethylene bridged hybrid (BEH) Amide Column, 130 Å, 1.7 μm, 2.1 mm×50 mm column (Waters, MA) was used to chromatographically separate N-glycans during the 18.3 min run employing a gradient of 50 mM Ammonium Formate pH 4.4 (Waters) made with LC-MS Water (Millipore), LC-MS ACN (VWR, Honeywell) 25%-75% gradient transitioning over 12 min to 60–40%. N-glycans separated by charge and stereochemistry were quantitated using Waters AQUITY Fluorescent detector set to 265/425 em/ex, 10 Hz using Empower 3 software. Lastly, N-glycan identity was confirmed using a Waters AQUITY QDa Mass spectrometer. The resulting UPLC fluorescent trace was analyzed with Empower v3.3.1 software, UPLC trace percent-area was combined with collected MS-spectra to identify eluted peaks as described previously[98]. Pooled N-glycans labeled with the RapiFluor tag were digested with Neuraminidase S (New England BioLabs, MA, P0743L) or Neuraminidase (New England BioLabs, MA, P0720S) for 12 hours at 32 °C following the manufacturer's instructions. Digested N-glycans were cleaned up using Water's SPE kit and analyzed using the UPLC detailed above.

### Antigen-specific complement deposition assay

Antibody-specific complement deposition against the RBD and spike S1 antigens were assayed following the previously developed protocol[69]. Briefly, 20 μL FluoSpheres™ NeutrAvidin™-Labeled Microspheres (ThermoFisher) were incubated with 20 μg RBD (aa319-541, Invitrogen) (biotinylated in-house using the EZ-Link™ Sulfo-NHS-LC-Biotinylation Kit) or 20 μg biotinylated SARS-CoV-2 (2019-nCoV) spike S1-His Recombinant Protein, Biotinylated (SinoBiological) antigen for 4 hours at 37 °C. After washing twice with 200 μL 1X PBS, the antigen-bound beads were blocked with 200 μL 5% BSA in 1X PBS for 1 hour at 37 °C. Next, the beads were washed twice with 500 μL of 0.1% BSA in 1X PBS and diluted 1:100 in 1X PBS. A subset of plasma and purified IgM samples were treated with a nonspecific neuraminidase (New England BioLabs, MA, P0720S) for 12 hours at 32 °C prior to antigen-specific complement deposition analysis following the manufacturer's instructions. Next, 15 μL of the 1:100 bead solution was transferred to low-binding 1.5 mL tubes (Corning) and incubated with 20 μL of 1:10 1X PBS diluted pooled severe or nonsevere plasma or 5 μg of IgM isolated from pooled severe or nonsevere plasma for 2 hours at 37 °C. Next, the immune-complexed beads were incubated for 15 minutes with freshly resuspended guinea pig complement (Cedarlane, CL4051) and diluted 1:50 in gelatin veronal buffer with Mg²⁺ & Ca²⁺ (GVB + +) at 37 °C. The complement deposition was halted with two washes of 200 μL 15 mM Ethylenediamine tetraacetic acid (EDTA). Next, 50 μL of a 1:100 diluted fluorescein isothiocyanate (FITC) labeled Goat anti-Guinea pig Complement C3 antibody (MP Biomedicals, 55385, lot #:08077) was incubated for 30 minutes with the immune-complexed beads. Lastly, two 200 μL 1X PBS washes removed unbound FITC labeled anti-C3 antibody. Washed samples were re-suspended in 100uL and analyzed using a Fortessa Flow Cytometer (BD). Beads were gated for the presence or absence of the FITC antibody, and the mean fluorescence intensity (MFI) of the bead content was divided by the total number of beads to determine the rate of complement deposition in each sample. The gating strategy is displayed in Fig. 4B. Flow Minus One (FMO) control samples were run with the same protocol to confirm a low background signal and inform the gating cut-off strategy.

### Statistical analysis

A biomarker was removed from analysis if its overall number of missing values was greater than 3 (13.6% of 22 patients) to reduce potential bias[99–101]. Data collection and analysis were performed using the software: Microsoft Excel v16.66.1, Waters Empower v3, R Studio v4.2.3, FacsDiva v9.0, and FlowJo v10.9. COVID-19 trajectory groups were categorized as nonsevere (trajectories 1-3) and severe (trajectories 4–5) cohorts for the measured transcriptomic, proteomic, Luminex, and clinical data. Sex and COVID-19 trajectories were summarized as counts and percentages. Continuous variables were summarized as the median and interquartile range (IQR) overall and by severity cohorts. For transcriptomic data, raw counts were normalized to counts per million (CPM) and then values were log2 transformed for statistical analysis. A pseudo-count of 2 was added to all CPM data prior to log transformation because zero cannot be 'logged'[102–104]. A Kruskal-Wallis test was used to test the significance of continuous variables between severe and nonsevere cohort clinical and laboratory data. Two-sided unpaired t-tests were used to compare two independent groups. One-way ANOVA with Tukey's multiple comparisons test was used to compare multiple group means. Associations between IgM mannosylated or total S2 and other variables were tested using simple linear regression. The coefficient of determination $R^2$ was obtained from linear regression. $p < 0.05$ was considered statistically significant for all tests.

### Reporting summary

Further information on research design is available in the Nature Portfolio Reporting Summary linked to this article.

## Data availability

All IMPACC data including those generated in this study have been deposited in the Immunology Database and Analysis Portal (ImmPort), a NIAID Division of Allergy, Immunology and Transplantation funded data repository under accession code SDY1760 (immport.org). All raw and processed data are available under restricted access to comply with the NIH public data sharing policy for IRB-exempted public health surveillance studies, Access can be obtained via AccessClinicalData@NIAID (https://accessclinicaldata.niaid.nih.gov/study-viewer/clinical_trials). Additional guidelines for access are outlined on ImmPort (https://docs.immport.org/home/impaccslides). The source data generated in the figures are provided as a Source Data file. Source data are provided with this paper.

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

## Acknowledgements

We acknowledge the patients who donated samples this study, medical students Brett Cohen, Nick Semenza, Brandon Rogowski, Sarah Fur-ukawa, Kristen Ulring, Nataliya Melnyk, and the Monobind AccuBind® ELISA Anti-SARS-CoV-2 IgG/IgM/IgA development team: Dr. Frederick Jerome, Christian Ayoub, Matthew Nguyen, and Anthony Shatola. The study was funded by the United States National Institutes of Health through the following grants: 5R01AI132774-03 (MCA), 5R01AI135803-03 (DBC, FK), 5U19AI118608-04 (OL, HS, LRB, JDA), 5U19AI128910-04 (EH, CBC, RPS, GAM), 4U19AI090023-11 (BP, NR, SB), 4U19AI118610-06 (VS, AFS, FK, HVB, SKS), R01AI145835-01A1S1 (WBM, CLH), 5U19AI062629-17 (JPM, NIAH), 5U19AI057229-17 (MMD, KCN, HM), 5U19AI125357-05 (MK, CB), 5U19AI128913-03 (JS, EFR), 3U19AI077439-13 (DE, CC, CRL, WE), 5U54AI142766-03 (MAA, SCB), 5R01AI104870-07 (LE, EM), and 3U19AI089992-09 (DH, RRM, ACS). The manuscript content is solely the responsibility of the authors and does not necessarily represent the official views of the NIH.

## Author contributions

B.H-G. contributed conceptualization, data curation, formal analysis, investigation, methodology, writing – original draft, and writing – review & editing. K.W. contributed data curation, formal analysis, visualization, and writing – review & editing. J.H. performed the statistical analysis and writing – review & editing. J.C., G.C., M.Bell, N.M., M.Bernui contributed data curation. S.F., M.C.A., F.K. and B.T. contributed data curation and formal analysis. N.R., J.D.A., H.B. and H.T.M. contributed by providing resources, investigation, and writing – review & editing. M.K., C.B.C. and B.W. provided resources and funding acquisition. E.K.H. and M.A.C contributed conceptualization, investigation, providing resources, funding acquisition, project administration, and writing – review & editing.

## Competing interests

The Icahn School of Medicine at Mount Sinai has filed patent applications relating to SARS-CoV-2 serological assays and NDV-based SARS-CoV-2 vaccines which list Florian Krammer (F.K.) as co-inventor. Mount Sinai has spun out a company, Kantaro, to market serological tests for SARS-CoV-2. F.K. has consulted for Merck, Seqirus, Curevac and Pfizer, and is currently consulting for GSK, Gritstone, 3rd Rock Ventures and Avimex and he is a co-founder and scientific advisory board member of CastleVax. The Krammer laboratory is also collaborating with Pfizer on animal models of SARS-CoV-2 and Dynavax on influenza virus vaccines. All other authors have no competing interests to declare.

## Additional information

## IMPACC Network

**IMPACC Steering Committee** Al Ozonoff[8], Joann Diray-Arce[8], Matthew C. Altman [3], Lauren I. R. Ehrlich[9], Esther Melamed[9], Ana Fernandez Sesma[4,5,6], Viviana Simon[4,5,6], Bali Pulendran[7], Kari C. Nadeau[7], Mark M. Davis[7], Grace A. McCoey[10], Rafick Sekaly[10], Charles B. Cairns [1], Elias K. Haddad [1]✉, Lindsey R. Baden[8], Ofer Levy[8], Joanna Schaenman[11], Elaine F. Reed[11], Albert C. Shaw[12], David A. Hafler[12], Ruth R. Montgomery[12], Steven H. Kleinstein[12], Nadine Rouphael [2], Patrice M. Becker[13], Alison D. Augustine[13], Carolyn S. Calfee[14], David J. Erle[14], Michael E. DeBakey[15], David B. Corry[16], Farrah Kheradmand[16], Mark A. Atkinson[17], Scott C. Brakenridge[17], Nelson I. Agudelo Higuita[16], Jordan P. Metcalf[16], Catherine L. Hough[18], William B. Messer[18], Monica Kraft[19], Chris Bime[19] & Bjoern Peters[20]

**Clinical & Data Coordinating Center (CDCC)** Al Ozonoff[8], Carly E. Milliren[8], Joann Diray-Arce[8], Caitlin Syphurs[8], Kerry McEnaney[8], Brenda Barton[8], Claudia Lentucci[8], Mehmet Saluvan[8], Ana C. Chang[8], Annmarie Hoch[8], Marisa Albert[8], Tanzia Shaheen[8], Alvin T. Kho[8], Shanshan Liu[8], Sanya Thomas[8], Jing Chen[8], Maimouna D. Murphy[8], Mitchell Cooney[8], Arash Nemati Hayati[8], Robert Bryant[8] & James Abraham[8]

**IMPACC Data Analysis Group** Al Ozonoff[8], Joann Diray-Arce[8], Naresh Doni Jayavelu[3], Matthew C. Altman [3], Scott Presnell[3], Tomasz Jancsyk[3], Cole Maguire[9], Jingjing Qi[4,5,6], Brian Lee[4,5,6], Slim Fourati[10], Charles B. Cairns [1], Denise A. Esserman[12], Leying Guan[12], Steven H. Kleinstein[12], Jeremy Gygi[12], Shrikant Pawar[12], Anderson Brito[12], Gabriela K. Fragiadakis[14], Ravi Patel[14], Michael E. DeBakey[15], Bjoern Peters[20], James A. Overton[20], Randi Vita[20], Kerstin Westendorf[20], Casey P. Shannon[21] & Scott J. Tebbutt[21]

**IMPACC Site Investigators** Lauren I. R. Ehrlich[9], Esther Melamed[9], Rama V. Thyagarajan[9], Justin F. Rousseau[9], Dennis Wylie[9], Todd A. Triplett[9], Erna Kojic[4,5,6], Viviana Simon[4,5,6], Kari C. Nadeau[7], Sharon Chinthrajah[7], Neera Ahuja[7], Angela J. Rogers[7], Maja Artandi[7], Linda Geng[7], Grace A. McCoey[10], George Yendewa[10], Charles B. Cairns[1], Debra L. Powell[1], James N. Kim[1], Brent Simmons[1], I. Michael Goonewardene[1], Cecilia M. Smith[1], Mark Martens[1], Lindsey R. Baden[8], Amy C. Sherman[8], Stephen R. Walsh[8], Nicolas C. Issa[8], Joanna Schaenman[11], Ramin Salehi-Rad[11], Albert C. Shaw[12], Charles Dela Cruz[12], Shelli Farhadian[12], Akiko Iwasaki[12], Albert I. Ko[12], Nadine Rouphael[2], Evan J. Anderson[2], Aneesh K. Mehta[2], Jonathan E. Sevransky[2], Vicki Seyfert-Margolis[22], Carolyn S. Calfee[14], David J. Erle[14], Aleksandra Leligdowicz[14], Michael A. Matthay[14], Jonathan P. Singer[14], Kirsten N. Kangelaris[14], Carolyn M. Hendrickson[14], Matthew F. Krummel[14], Charles R. Langelier[14], Prescott G. Woodruff[14], David B. Corry[14], Farrah Kheradmand[14], Scott C. Brakenridge[17], Matthew L. Anderson[17], Faheem W. Guirgis[17], Nelson I. Agudelo Higuita[16], Jordan P. Metcalf[16], Douglas A. Drevets[16], Brent R. Brown[16], William B. Messer[18], Sarah A. R. Siegel[18], Zhengchun Lu[18], Monica Kraft[19], Chris Bime[19], Jarrod Mosier[19] & Hiroki Kimura[19]

**IMPACC Core Laboratory** Joann Diray-Arce[8], Matthew C. Altman[3], Bernard Khor[3], Florian Krammer[4,5,6], Harm van Bakel[4,5,6], Adeeb Rahman[4,5,6], Daniel Stadlbauer[4,5,6], Jayeeta Dutta[4,5,6], Hui Xie[4,5,6], Seunghee Kim-Schulze[4,5,6], Ana Silvia Gonzalez-Reiche[4,5,6], Adriana van de Guchte[4,5,6], Juan Manuel Carreño[4,5,6], Gagandeep Singh[4,5,6], Ariel Raskin[4,5,6], Johnstone Tcheou[4,5,6], Dominika Bielak[4,5,6], Hisaaki Kawabata[4,5,6], Brian Lee[4,5,6], Geoffrey Kelly[4,5,6], Manishkumar Patel[4,5,6], Hui Xie[4,5,6], Kai Nie[4,5,6], Temima Yellin[4,5,6], Miriam Fried[4,5,6], Leeba Sullivan[4,5,6], Sara Morris[4,5,6], Holden T. Maecker[7], Scott Sieg[10], Hanno Steen[8], Patrick van Zalm[8], Benoit Fatou[8], Kevin Mendez[8], Jessica Lasky-Su[8], Scott R. Hutton[8], Greg Michelotti[8], Kari Wong[8], Meenakshi Jha[8], Arthur Viode[8], Naama Kanarek[8], Boryana Petrova[8], Albert C. Shaw[12], Yujiao Zhao[12], Charles Dela Cruz[12], Ruth R. Montgomery[12], Steven E. Bosinger[2], Arun K. Boddapati[2], Greg K. Tharp[2], Kathryn L. Pellegrini[2], Elizabeth Beagle[2], David Cowan[2], Sydney Hamilton[2], Susan Pereira Ribeiro[2], Thomas Hodder[2], Lindsey B. Rosen[13], Serena Lee[13], Charles R. Langelier[14], Michael R. Wilson[14], Ravi Dandekar[14], Bonny Alvarenga[14], Jayant Rajan[14], Walter Eckalbar[14], Andrew W. Schroeder[14], Alexandra Tsitsiklis[14], Eran Mick[14], Yanedth Sanchez Guerrero[14], Christina Love[14], Lenka Maliskova[14] & Michael Adkisson[14]

**IMPACC Clinical Study Team** Cole Maguire[9], Nadia Siles[9], Janelle Geltman[9], Kerin Hurley[9], Miti Saksena[4,5,6], Deena Altman[4,5,6], Erna Kojic[4,5,6], Komal Srivastava[4,5,6], Lily Q. Eaker[4,5,6], Maria C. Bermúdez-González[4,5,6], Katherine F. Beach[4,5,6], Levy A. Sominsky[4,5,6], Arman R. Azad[4,5,6], Lubbertus C. F. Mulder[4,5,6], Giulio Kleiner[4,5,6], Alexandra S. Lee[7], Evan Do[7], Andrea Fernandes[7], Monali Manohar[7], Thomas Hagan[7], Catherine A. Blish[7], Hena Naz Din[7], Jonasel Roque[7], Samuel Yang[7], Natalia Sigal[7], Iris Chang[7], Heather Tribout[10], Paul Harris[10], Mary Consolo[10], Mariana Bernui[1], Michele A. Kutzler[1], Carolyn Edwards[1], Jennifer Connors[1], Edward Lee[1], Edward Lin[1], Brett Croen[1], Nicholas C. Semenza[1], Brandon Rogowski[1], Nataliya Melnyk[1], Kyra Woloszczuk[1], Gina Cusimano[1], Mathew R. Bell[1], Sara Furukawa[1], Renee McLin[1], Pamela Schearer[1], Julie Sheidy[1], George P. Tegos[1], Crystal Nagle[1], Ofer Levy[8], Kinga Smolen[8], Michael Desjardins[8], Simon van Haren[8], Xhoi Mitre[8], Jessica Cauley[8], Xiaofang Li[8], Alexandra Tong[8], Bethany Evans[8], Christina Montesano[8], Jose Humberto Licona[8], Jonathan Krauss[8], Jun Bai Park Chang[8], Natalie Izaguirre[8], Rebecca Rooks[8], David Elashoff[11], Jenny Brook[11], Estefania Ramires-Sanchez[11], Megan Llamas[11], Adreanne Rivera[11], Claudia Perdomo[11], Dawn C. Ward[11], Clara E. Magyar[11], Jennifer A. Fulcher[11], Harry C. Pickering[11], Subha Sen[11], Omkar Chaudhary[12], Andreas Coppi[12], John Fournier[12], Subhasis Mohanty[12], Catherine Muenker[12], Allison Nelson[12], Khadir Raddassi[12], Michael Rainone[12], William E. Ruff[12], Syim Salahuddin[12], Wade L. Schulz[12], Pavithra Vijayakumar[12], Haowei Wang[12], Elsio Wunder Jr.[12], H. Patrick Young[12], Yujiao Zhao[12], Jessica Rothman[12], Anna Konstorum[12], Ernie Chen[12], Chris Cotsapas[12], Nathan D. Grubaugh[12], Xiaomei Wang[12], Leqi Xu[12], Hiromitsu Asashima[12], Laurel Bristow[2], Laila Hussaini[2], Kieffer Hellmeister[2], Hady Samaha[2], Sonia Tandon Wimalasena[2], Andrew Cheng[2], Christine Spainhour[2], Erin M. Scherer[2], Brandi Johnson[2], Amer Bechnak[2], Caroline R. Ciric[2], Lauren Hewitt[2], Erin Carter[2], Nina Mcnair[2], Bernadine Panganiban[2], Christopher Huerta[2], Jacob Usher[2], Vicki Seyfert-Margolis[22], Tatyana Vaysman[13], Steven M. Holland[13], Yumiko Abe-Jones[14], Saurabh Asthana[14], Alexander Beagle[14], Sharvari Bhide[14], Sidney A. Carrillo[14], Suzanna Chak[14], Gabriela K. Fragiadakis[14], Rajani Ghale[14], Ana Gonzalez[14], Alejandra Jauregui[14], Norman Jones[14], Tasha Lea[14], Deanna Lee[14], Raphael Lota[14], Jeff Milush[14], Viet Nguyen[14], Logan Pierce[14], Priya A. Prasad[14], Arjun Rao[14], Bushra Samad[14], Cole Shaw[14], Austin Sigman[14], Pratik Sinha[14], Alyssa Ward[14], Andrew Willmore[14], Jenny Zhan[14], Sadeed Rashid[14], Nicklaus Rodriguez[14], Kevin Tang[14], Luz Torres Altamirano[14], Legna Betancourt[14], Cindy Curiel[14], Nicole Sutter[14], Maria Tercero Paz[14], Gayelan Tietje-Ulrich[14], Carolyn Leroux[14], Ravi Patel[14], Neeta Thakur[14], Joshua J. Vasquez[14], Lekshmi Santhosh[14], Li-Zhen Song[15], Ebony Nelson[15], Lyle L. Moldawer[17], Brittany Borresen[17], Brittney Roth-Manning[17], Ricardo F. Ungaro[17], Jordan Oberhaus[17], J. Leland Booth[16],

Lauren A. Sinko[16], Amanda Brunton[18], Peter E. Sullivan[18], Matthew Strnad[18], Zoe L. Lyski[18], Felicity J. Coulter[18], Courtney Micheleti[18], Michelle Conway[19], Dave Francisco[19], Allyson Molzahn[19], Heidi Erickson[19], Connie Cathleen Wilson[19], Ron Schunk[19], Bianca Sierra[19] & Trina Hughes[19]

[9]The University of Texas at Austin, Austin, TX, USA. [10]Case Western Reserve University and University Hospitals of Cleveland, Cleveland, OH, USA. [11]David Geffen School of Medicine at the University of California Los Angeles, Los Angeles, CA, USA. [12]Yale School of Medicine, and Yale School of Public Health, New Haven, CT, USA. [13]National Institute of Allergy and Infectious Diseases/National Institutes of Health, Bethesda, MD, USA. [14]University of California San Francisco School of Medicine, San Francisco, CA, USA. [15]Baylor College of Medicine, and the Center for Translational Research on Inflammatory Diseases, Michael E. DeBakey VA Medical Center, Houston, TX, USA. [16]Oklahoma University Health Sciences Center, Oklahoma City, OK, USA. [17]University of Florida/ University of South Florida, Tampa, FL, USA. [18]Oregon Health & Science University, Portland, OR, USA. [19]University of Arizona, Tucson, AZ, USA. [20]La Jolla Institute for Immunology, La Jolla, CA, USA. [21]Centre for Heart Lung Innovation, Providence Research, St. Paul's Hospital, and the PROOF Centre of Excellence, Vancouver, BC, Canada. [22]MyOwnMed, Inc, Bethesda, MD, USA.

