## [Peer Review File · Nature Communications]

IgM N-glycosylation correlates with COVID-19 severity and rate of complement depositionReviewers' Comments:

Reviewer #1:

Remarks to the Author:

Haslund-Gourley and colleagues have assessed IgM N-glycosylation patterns in the setting of early acute COVID-19 in a cohort of 22 hospitalized patients, of which 10 had a severe and 12 a non-severe disease course. They found and convincingly show differences in IgM mannose and di-sialylated (S2) N-glycans. Moreover, via a bulk RNA sequencing dataset of the same patients, the authors can link these phenotypes to reduced glycosyltransferase/glycosidase mRNA expression in PBMCs. However, further associations with clinical and serological markers are hindered by the low number of observations, making the interpretation of associated markers such as IL-18 difficult. The final complement activation assays give more mechanistic insights, as the increased complement activation elicited by purified IgM from severe patients, compared to non-severe patients, was abrogated by the addition of mannosidase or sialidase. This observation further increases the potential relevance of their finding on N-glycosylation patterns of IgM in severe COVID-19 patients. Overall, I found this study well reported.

Major points:

Figure 3: The glycosyltransferase expression levels appear to be analyzed in all PBMCs and not only B cells/plasmablasts. Can the authors look at only the B cells/ plasmablasts in this dataset?

Figure 4: The authors strengthen their finding of changes in IgM N-glycosylation and glycosyltransferase expression by correlating the glycosylation patterns with serological correlates of COVID-19 severity. However, the low patient numbers appear to affect the statistical significances, particularly shown by the observation that between the non-severe and the severe patients, only the cytokine IL-18 is elevated as reported in line 549 and Supplementary Table 4. The interpretation that IL-16 and IL-18 "may play a role in controlling downstream glycosyltransferase expression" is questionable based on the presented data. Can the authors provide further evidence for the suggested mechanism? Does for example a linear model correcting for patient age, sex, and severity confirm the association of IL-16/18 with the glycosyltransferase expression levels?

Figure 5: The finding of increased complement activation in severe COVID-19 patients by 5ug purified IgM is very interesting. Can differences in the amount of IgM binding to the beads potentially lead to differences in subsequent complement activation, e.g. by different affinities to SARS-CoV-2 spike in the severe, compared to non-severe?

Also, do the authors see correlations between the complement activation capacity and total mannose or S2 on a patient level? Can the abrogation of this effect by mannosidase or sialidase be quantified and correlated to the total mannose or S2 on a patient level?

Minor points:

Table 1: The comparison across the 5 groups is difficult by eye. Can the authors add a table merging the non-severe patients and severe patients respectively (similar to Suppl. Table 4 and most figures of the manuscript). Also, please show already in the main table D-Dimer, BUN, and Creatinine measurements as continuous variables with median + interquartile range, instead of as categorical variables, which makes comparisons difficult. Age, BMI, symptom onset to hospitalization days, and SOFA score would also profit from such reporting here.

Figure 1B: The groups are very well balanced regarding the clinical characteristics, despite having very different COVID-19 severities (one could expect older, and more male patients in group 5). Have the authors matched the trajectory groups selected for this study previous to the experimental exploration? Were there any inclusion criteria or potential biases for the 22 patients in this study deriving from a cohort with 1164 patients according to the cited paper by Ozonoff et al.

([https://www.thelancet.com/journals/ebiom/article/PIIS2352-3964\(22\)00390-5](https://www.thelancet.com/journals/ebiom/article/PIIS2352-3964(22)00390-5))

Figure 1C is well presented and convincingly shows the differences within the patient subgroups.

Lines 170-180: Please also consider citing the following key paper by Chakraborty et al.

(<https://www.science.org/doi/10.1126/scitranslmed.abm7853>), who additionally show differences in antibody fucosylation between SARS-CoV-2 infected and COVID-19 vaccinated patients.

Between which months were the patients included in this study? Please report in the methods section.

The lack of vaccination could already mentioned in the methods section on the human subjects.

Line 196: The authors introduce the ability of IgM to activate the complement cascade by both the classical and the lectin pathways. Here they cite Hiatt et al., which is a commentary on the engineering of IgM in plants, without mentioning lectin pathway activation neither in the commentary nor the associated original article. The second citation, Zhang et al., shows is based on a natural IgM in a mouse model.

Can the authors expand this part of the introduction by also considering data in humans (e.g. [https://www.jbc.org/article/S0021-9258\(20\)56470-3](https://www.jbc.org/article/S0021-9258(20)56470-3))? This could benefit the later interpretation of increased complement activation.

Lines 539 -544: Please provide also a direct comparison of SARS-CoV-2-specific antibody levels between the analyzed patient trajectory groups (boxplots).

Discussion: Do the authors have access to clinical follow-up data on the 22 patients? As total IgM has been associated with long Covid (<https://www.nature.com/articles/s41467-021-27797-1>), it may be interesting to correlate the N-glycosylation with this pattern or discuss the potential relevance of IgM mannose and S2 also in the post-acute setting.

Reviewer #2:

Remarks to the Author:

The study unveils an interesting trend in IgM N-glycosylation in COVID-19 patient cohorts and expands on the current understanding of IgG and IgM on disease states. The study design and analysis are sound, and the presentation is acceptable or adequate. However, the significance of the study is downplayed by the correlational nature of the study, therefore a direct cause-consequence relationship is not clear.

General comments:

1. The bar patterns in Figure 2 are not very differentiating between Day 4 and Day 7. Please use a more distinguishable pattern/color.
2. Red dash in Figure 5E figure title should be omitted.

Significant comments:

1. Since the study predominantly focuses on differential glycosylation between Ig serotypes, after IgG / IgM isolation, the authors should assess protein purity and successful purification – e.g. from MS or PAGE.
2. The authors state IgM glycosylation is more impactful on immune regulation. However, if the trend in sialylation is opposite in trends to IgG, how do the authors rule out whether the trends in immunogenicity are due to IgG or IgM – particularly because the immunogenicity trends the authors mention correlate to established sialylation characteristics in IgG? Specifically, in Figures 2 and 3, the authors show glycosyltransferase expression differs in the two trajectories (1-3, 4-5), whereas only

IgM glycosylation shows a significant difference. The authors should also show a similar categorization by disease trajectory for Figure 4 to justify their statement about IgM glycosylation.

3. Related to Q2, outside of correlational data, the study shows direct evidence of the immunogenic response of IgM only in the last figure. The argument for the glycosylation/sialylation effect resulting for IgM would have been stronger if binding/activity assays would be implemented with purified IgM.

4. In the manuscript abstract figure, the authors hypothesized IL-18 and IL-16 might alter glycosyltransferase expression levels. The figure shows a question mark indicating the entire picture is not complete. Alternatively, can the authors show a linkage between IL-18/IL-16 addition/induction and glycosyltransferase expression in a model cell line?

5. The authors should show a curve of mannosidase and sialidase-digested IgM to show the specific glycoforms analyzed in Figure 5E – this relates to Figure 2C where the authors show IgM M4-M6 and M7-M10 are significantly regulated oppositely. Which category is more impactful?

6. What is the author's interpretation of Figure 5C-E? In Figure 5E, it appears IgM is not responsive to complement deposition in phases 1-3. Therefore, the complement activation response in plasma is likely from other plasma components or Ig serotypes – which is consistently observed in states 1-3 and 4-5. On the other hand, with severe disease states (4-5), sialidase treatment had a strong effect on IgM complement deposition but did not seem to be responsive in complement activation in plasma. Don't these observations indicate that N-glycosylation of IgM is not responsible for complement deposition in plasma, but other components/serotypes? Please clarify in manuscript text.

7. Related to Q6, did the authors measure the percentage of IgM serotypes compared to total Ig?

8. Related to Q2&6, the authors should show IgG N-glycosylation correlation statistics as a supplementary figure or table to show the significance of IgM, and not IgG, glycosylation on clinical and transcriptomic parameters.

Reviewer #3:

Remarks to the Author:

The paper of Haslund-Gourley et al describes the changes in IgM (and G) total galactosylation in the first few weeks after a naïve SARS-CoV-2 infection. While interesting, to my knowledge IgM has not been analyzed in COVID-19 before, the study suffer tremendously from previous publication of specific IgG glycosylation. The relevance looking only at specific IgG glycosylation is perhaps best exemplified in figure 2, where the most severe individuals show lowered total IgG fucosylation, that is reminiscent to what has been published before, and is most likely the result of the massive amount of specific IgG produced that is transiently afucosylated. The changes in IgG galactosylation in the most severe individuals has also been described before (very strong reduction of both specific (from 80 to 20% but also total IgG galactosylation seen in the most severe). None of the studies looking at antigen-specific (and total) IgG glycosylation in this infection is introduced as context nor discussed. Remarkably, this study shows that this is not reflected in total IgM. However, the lack of specific Ig analysis makes this study less relevant.

Major

-The methods describing glycosyltransferase expression seems to be missing

-The rationale to link total IgM glycan features to the amount of IgA/M and G produce to anti-N is absent. What is the hypothesis?

-Epitopes of S1 and RBD are largely overlapping, rationale missing, also why not S2, or why full spike was not used. The reason why S1 and RBD are so different is likely to be a technical problem.

Furthermore, using plasma also contains IgG which also activates complement. Seems to be

compensated in D and E, but conclusion in the results and figure legends are blurry and misleading.
-Discussion is long and not always relevant (and even off topic).

minor

Results/Figure 1: GP not defined.

Fig. 1B, please show also individual points in B.

Fig. 1C, the patterns for day 4 and 7 are not clearly distinguishable.

Fig 5A middle, it is not clear that you are showing beads here, looks like cells. Please make this clear.

Fig. 5A right IgM is shown as a symmetric pentamer by has actually a hexameric configuration with the J chain occupying one of the hexagonal space. It is not clear where the information about the relationship between location and type of glycan in IgM comes from.

REVIEWER COMMENTS

Reviewer #1 (Remarks to the Author):

Haslund-Gourley and colleagues have assessed IgM N-glycosylation patterns in the setting of early acute COVID-19 in a cohort of 22 hospitalized patients, of which 10 had a severe and 12 a non-severe disease course. They found and convincingly show differences in IgM mannose and di-sialylated (S2) N-glycans. Moreover, via a bulk RNA sequencing dataset of the same patients, the authors can link these phenotypes to reduced glycosyltransferase/glycosidase mRNA expression in PBMCs. However, further associations with clinical and serological markers are hindered by the low number of observations, making the interpretation of associated markers such as IL-18 difficult.

The final complement activation assays give more mechanistic insights, as the increased complement activation elicited by purified IgM from severe patients, compared to non-severe patients, was abrogated by the addition of mannosidase or sialidase. This observation further increases the potential relevance of their finding on N-glycosylation patterns of IgM in severe COVID-19 patients. Overall, I found this study well-reported.

Thank you, we appreciate your insight into the limitation of the associations of the IgM N-glycan profile with serological markers and clinical laboratory data. We have revised **Figure 4** to reflect a more conservative approach to the correlations to IgM N-glycans.

Major points:

Figure 3: The glycosyltransferase expression levels appear to be analyzed in all PBMCs and not only B cells/plasmablasts. Can the authors look at only the B cells/ plasmablasts in this dataset?

We would have preferred to analyze B cell/plasmablast-specific glycosyltransferase expression, unfortunately, our collaborators only analyzed total PBMC mRNA expression. We note this important distinction in the results section and elaborated upon this fact in the results and discussion section of the revised manuscript.

Figure 4: The authors strengthen their finding of changes in IgM N-glycosylation and glycosyltransferase expression by correlating the glycosylation patterns with serological correlates of COVID-19 severity. However, the low patient numbers appear to affect the statistical significance, particularly shown by the observation that between the non-severe and the severe patients, only the cytokine IL-18 is elevated as reported in line 549 and Supplementary Table 4. The interpretation that IL-16 and IL-18 “may play a role in controlling downstream glycosyltransferase expression” is questionable based on the presented data. Can the authors provide further evidence for the suggested mechanism? Does for example a linear model correcting for patient age, sex, and severity confirm the association of IL-16/18 with the glycosyltransferase expression levels?

We agree that our interpretation of cytokines IL-16 and IL-18 impacting IgM N-glycosylation does not meet the threshold to claim correlation. We have removed IL-16 and IL-18 from **Figure 4** and updated the results and discussion sections to reflect a more conservative consideration of these cytokines.

Figure 5: The finding of increased complement activation in severe COVID-19 patients by 5ug purified IgM is very interesting. Can differences in the amount of IgM binding to the beads potentially lead to differences in subsequent complement activation, e.g. by different affinities to SARS-CoV-2 spike in the severe, compared to non-severe?

The difference in the amount of IgM binding to the antigen-coated beads could lead to differences in subsequent complement activation. We have updated the results and discussion to address this point in our revised manuscript.

Also, do the authors see correlations between the complement activation capacity and total mannose or S2 on a patient level? Can the abrogation of this effect by mannosidase or sialidase be quantified and correlated to the total mannose or S2 on a patient level?

These are excellent questions, and this would be a fantastic next step for the project, however, at this time we did not have sufficient plasma to explore on an individual level. To complete these studies with the available allocated plasma, we created pools of severe and nonsevere cohorts rather than run individual samples from the five trajectories. Future studies should examine the relationship between S2 and mannose content on an individual patient level. We have updated our future directions to speak to this point.

Minor points:

Table 1: The comparison across the 5 groups is difficult by eye. Can the authors add a table merging the non-severe patients and severe patients respectively (similar to Suppl. Table 4 and most figures of the manuscript).

We have updated Table 1 with two additional columns reflecting the nonsevere or severe cohort grouping to increase the ease of comparison.

Also, please show already in the main table D-Dimer, BUN, and Creatinine measurements as continuous variables with median + interquartile range, instead of as categorical variables, which makes comparisons difficult. Age, BMI, symptom onset to hospitalization days, and SOFA score would also profit from such reporting here.

We have updated Table 1 to present Age, D-Dimer, BUN, Creatinine, and SOFA score as continual variables with mean +/- Standard Deviation (S.D.). However, we feel that BMI and symptom onset to hospitalization days provide more detailed information to the reader remaining in their current format.

Figure 1B: The groups are very well balanced regarding the clinical characteristics, despite having very different COVID-19 severities (one could expect older, and more male patients in group 5). Have the authors matched the trajectory groups selected for this study previous to the experimental exploration?

Patient samples were selected to offer balanced clinical characteristics (sex, BMI, age) prior to experimental assays. Trajectories were determined in the Ozonoff et al. publication. We have revised the human sample methods section to communicate these points.

Were there any inclusion criteria or potential biases for the 22 patients in this study deriving from a cohort with 1164 patients according to the cited paper by Ozonoff et al.

([https://www.thelancet.com/journals/ebiom/article/PIIS2352-3964\(22\)00390-5](https://www.thelancet.com/journals/ebiom/article/PIIS2352-3964(22)00390-5))

These samples were obtained from the Drexel IMPACC plasma biorepository, with the goal of balancing the male and female sexes and the number of patients in the severe vs nonsevere cohorts. There were no other inclusion criteria or potential biases for the 22 patients in this study.

Figure 1C is well presented and convincingly shows the differences within the patient subgroups.

Lines 170-180: Please also consider citing the following key paper by Chakraborty et al. (<https://www.science.org/doi/10.1126/scitranslmed.abm7853>), who additionally show differences in antibody fucosylation between SARS-CoV-2 infected and COVID-19 vaccinated patients.

Thank you, we have updated the manuscript to include this citation on line 203.

Between which months were the patients included in this study? Please report in the methods section.

We have updated the methods section and introduction to inform readers that these sera samples were collected between the months of May 2020 through December 2020.

The lack of vaccination could already mentioned in the methods section on the human subjects.

We have updated the methods section to detail the lack of vaccinations available for the patient populations due to the early stage of the pandemic.

Line 196: The authors introduce the ability of IgM to activate the complement cascade by both the classical and the lectin pathways. Here they cite Hiatt et al., which is a commentary on the engineering of IgM in plants, without mentioning lectin pathway activation neither in the commentary nor the associated original article. The second citation, Zhang et al., shows is based on a natural IgM in a mouse model. Can the authors expand this part of the introduction by also considering data in humans (e.g. [https://www.jbc.org/article/S0021-9258\(20\)56470-3](https://www.jbc.org/article/S0021-9258(20)56470-3))? This could benefit the later interpretation of increased complement activation.

We apologize for the lack of clarity on these citations. We removed the plant IgM engineering commentary article, and instead support the claim of IgM N-glycans interacting with MBL and the lectin complement pathway citing the Arnold et al. 2005 (human), and Zhang et al. 2006 (murine model). Further we cite Sharp et al. 2019 in support of IgM N-glycan's role in classical complement deposition.

This section now reads: *"In addition, evidence supports IgM N-glycans interacting with C1q in the classical complement pathway and the mannan-binding lectin (MBL) associated with the lectin pathway of complement activation."*

Lines 539 -544: Please provide also a direct comparison of SARS-CoV-2-specific antibody levels between the analyzed patient trajectory groups (boxplots).

We recognize the importance of presenting severe and nonsevere patient SARS-CoV-2 specific antibody levels. We present these data in **Supplemental Table 7** and include associated p-values from student's T-test. We have updated the results section to direct interested readers to **Supplemental Table 7** to assess the SARS-CoV-2 specific antibody levels.

Discussion: Do the authors have access to clinical follow-up data on the 22 patients? As total IgM has been associated with long Covid (<https://www.nature.com/articles/s41467-021-27797-1>), it may be interesting to correlate the N-glycosylation with this pattern or discuss the potential relevance of IgM mannose and S2 also in the post-acute setting.

We agree this is an important point to make during the discussion. We have varying degrees of access to clinical follow-up to the patients in this study, but this study was not designed to capture data on long-COVID-19. We have updated our discussion and cite the suggested article in our revised manuscript.

Reviewer #2 (Remarks to the Author):

The study unveils an interesting trend in IgM N-glycosylation in COVID-19 patient cohorts and expands on the current understanding of IgG and IgM on disease states. The study design and analysis are sound, and the presentation is acceptable or adequate. However, the significance of the study is downplayed by the correlational nature of the study, therefore a direct cause-consequence relationship is not clear.

General comments:

1. The bar patterns in Figure 2 are not very differentiating between Day 4 and Day 7. Please use a more distinguishable pattern/color.

Thank you, we have updated **Fig 1C** to better differentiate the Day 4 and Day 7 using large white stripes that more easily visualized.

2. Red dash in Figure 5E figure title should be omitted.

We have omitted the red dash under this figure title (which is now **Figure 6**).

Significant comments:

1. Since the study predominantly focuses on differential glycosylation between Ig serotypes, after IgG / IgM isolation, the authors should assess protein purity and successful purification – e.g. from MS or PAGE.

Thank you, our purification of the total plasma IgG and IgM was confirmed via western blot and SDS-PAGE gels. We have revised the manuscript to include SDS-PAGE Coomassie-stained gels and western blot confirmation of total IgG and total IgM proteins enriched from total plasma in **Supplemental Figures 6A and 6B**.

In addition, we assessed the enrichment of the Spike S1-specific IgG and IgM for LC-MS/MS analysis and include the results of this analysis in **Supplemental Figures 6C and 6D**. We acknowledge there is a low level of IgG detected in the 75kDa band. This does not present a concern as IgM contains 5-times the number of glycosylation sites compared to IgG and reassuringly, the most abundant IgG N-glycan, FA2G0, constitutes only 0.5-0.6% of the IgM N-glycan profile. Thus, any N-glycans from IgG are negligible and do not contribute to the changes we observe within the IgM N-glycan profile.

2. The authors state IgM glycosylation is more impactful on immune regulation. However, if the trend in sialylation is opposite in trends to IgG, how do the authors rule out whether the trends in immunogenicity are due to IgG or IgM – particularly because the immunogenicity trends the authors mention correlate to established sialylation characteristics in IgG?

We do not intend to claim that IgM N-glycosylation is more impactful on immune regulation compared to the role that IgG N-glycans play. Rather, we seek to highlight the potential roles of IgM N-glycans during severe COVID-19. We have revised the manuscript to clarify this point.

Specifically, in Figures 2 and 3, the authors show glycosyltransferase expression differs in the two trajectories (1-3, 4-5), whereas only IgM glycosylation shows a significant difference. The authors should also show a similar categorization by disease trajectory for Figure 4 to justify their statement about IgM glycosylation.

This is an important point, we have included **Supplementary Tables 4, 5, and 6** in the revised manuscript to assess IgG agalactosylation, di-galactosylation, and mono-sialylation correlations with the glycosyltransferase and glycosidase transcriptomic dataset.

3. Related to Q2, outside of correlational data, the study shows direct evidence of the immunogenic response of IgM only in the last figure. The argument for the glycosylation/sialylation effect resulting for IgM would have been stronger if binding/activity assays would be implemented with purified IgM.

We agree that data connecting the glycosylation of IgM to the immunogenic response of IgM-dependent ADCD required further experimentation. During our revisions, we isolated SARS-CoV-2 Spike S1-specific IgG and IgM from severe and nonsevere COVID-19 cohorts (**Figure 5**). This analysis revealed that Spike S1-specific IgG N-glycosylation presented with a highly pro-inflammatory glycan phenotype, with elevated levels of FA2G0 in the severe COVID-19 cohort. On the other hand, Spike S1-specific IgM N-glycosylation contained elevated levels of sialic acid and high-mannose (M7-M10) moieties in the severe COVID-19 cohort compared to the nonsevere cohort. We believe that the antigen-specific IgM N-glycosylation analysis better supports the observed differences between the severe and nonsevere IgM-dependent ADCD presented in **Figure 6**.

4. In the manuscript abstract figure, the authors hypothesized IL-18 and IL-16 might alter glycosyltransferase expression levels. The figure shows a question mark indicating the entire picture is not complete. Alternatively, can the authors show a linkage between IL-18/IL-16 addition/induction and glycosyltransferase expression in a model cell line?

We thank the reviewer for this comment and agree that we do not present enough evidence in support of IL-18 and IL-16 to directly impact the glycosylation of IgM. We revised our manuscript to downplay the potential role of IL-18 and IL-16.

5. The authors should show a curve of mannosidase and sialidase-digested IgM to show the specific glycoforms analyzed in Figure 5E – this relates to Figure 2C where the authors show IgM M4-M6 and M7-M10 are significantly regulated oppositely. Which category is more impactful?

This is a great question; it is hard to say which (mannose or sialic acid content) is the more impactful N-glycosylation trend observed on IgM isolated from the severe COVID-19 cohort. The sialidase digestion in **Figure 6D** suggests sialic acid plays a significant role in IgM complement deposition. We have updated the manuscript to discuss these points.

6. What is the author's interpretation of Figure 5C-E? In Figure 5E, it appears IgM is not responsive to complement deposition in phases 1-3. Therefore, the complement activation response in plasma is likely from other plasma components or Ig serotypes – which is consistently observed in states 1-3 and 4-5. On the other hand, with severe disease states (4-5), sialidase treatment had a strong effect on IgM complement deposition but did not seem to be responsive in complement activation in plasma. Don't these observations indicate that N-glycosylation of IgM is not responsible for complement deposition in plasma, but other components/serotypes? Please clarify in manuscript text.

Thank you. We agree that in total plasma, the deposition of complement in an antigen-specific manner is predominantly promoted by IgG. This is because IgG is highly abundant and may have a higher affinity to the presented antigen. We have clarified this point in the revised manuscript.

Nevertheless, purified IgM also induces complement deposition in an antigen-specific manner, and that exoglycosidase digestions of the purified IgM modulate the ability of Spike S1-specific IgM to deposit complement. This suggests that IgM contributes to the overall rate of complement deposition in COVID-19 patients and may promote more complement deposition in the severe COVID-19 cohort. We have revised the manuscript to clarify these points as well.

7. Related to Q6, did the authors measure the percentage of IgM serotypes compared to total Ig?

During our revision, we isolated Spike S1-specific immunoglobulins from severe and nonsevere COVID-19 cohorts and analyzed the resulting abundance of the Spike S1-specific IgG (50kDa), and IgM (75kDa) using a Coomassie-stained SDS-PAGE gel followed by LC-MS/MS presented in **Supplementary Figures 6C and 6D**. There, we observed a higher abundance of the IgM heavy chain in the severe COVID-19 cohort compared to the nonsevere cohort. We have revised the manuscript to include this finding.

8. Related to Q2&6, the authors should show IgG N-glycosylation correlation statistics as a supplementary figure or table to show the significance of IgM, and not IgG, glycosylation on clinical and transcriptomic parameters.

This is an excellent point. We revised the manuscript to include **Supplementary Tables 4, 5, and 6** which compares the IgG N-glycosylation that significantly differed between the severe and nonsevere cohorts (G0, G2, and S1). We have cited this table in the results section to contrast the findings of IgM to those of IgG N-glycosylation. In brief, the IgG N-glycosylation data did not correlate with BUN, creatinine, potassium, or anti-N antibody abundance. However, IgG and IgM N-glycosylation trends correlated with elevated D-dimer and decreased MAN1A2 expression in

the severe cohort. We have revised the manuscript to highlight these results in the results and discussion sections.

Reviewer #3 (Remarks to the Author):

The paper of Haslund-Gourley et al describes the changes in IgM (and G) total galactosylation in the first few weeks after a naïve SARS-CoV-2 infection. While interesting, to my knowledge IgM has not been analyzed in COVID-19 before, the study suffers tremendously from previous publication of specific IgG glycosylation. The relevance of looking only at specific IgG glycosylation is perhaps best exemplified in figure 2, where the most severe individuals show lowered total IgG fucosylation, that is reminiscent to what has been published before, and is most likely the result of the massive amount of specific IgG produced that is transiently afucosylated. The changes in IgG galactosylation in the most severe individuals has also been described before (very strong reduction of both specific (from 80 to 20% but also total IgG galactosylation seen in the most severe). None of the studies looking at antigen-specific (and total) IgG glycosylation in this infection is introduced as context nor discussed.

We acknowledge the reviewer's important point about examining the antigen-specific immunoglobulin glycosylation, However, we respectfully disagree with their assertion that "None of the studies looking at antigen-specific (and total) IgG glycosylation in this infection is introduced as context nor discussed."

In the introduction of our manuscript, we cite and discuss sources (below) demonstrating previous IgG N-glycosylation studies during severe COVID-19:

- Pongracz, T., et al., *Immunoglobulin G1 Fc glycosylation as an early hallmark of severe COVID-19*. EBioMedicine, 2022. **78**: p. 103957.
- Hou, H., et al., *Profile of Immunoglobulin G N-Glycome in COVID-19 Patients: A Case-Control Study*. Frontiers in Immunology, 2021. **12**.
- Kljaković-Gašpić Batinjan, M., et al., *Differences in Immunoglobulin G Glycosylation Between Influenza and COVID-19 Patients*. Engineering (Beijing), 2022.
- Petrović, T., et al., *IgG N-glycome changes during the course of severe COVID-19: An observational study*. eBioMedicine, 2022. **81**: p. 104101.
- Vicente, M.M., et al., *Altered IgG glycosylation at COVID-19 diagnosis predicts disease severity*. European Journal of Immunology, 2022. **52**(6): p. 946-957.
- Chakraborty, S., et al., *Early non-neutralizing, afucosylated antibody responses are associated with COVID-19 severity*. Science Translational Medicine, 2022. **14**(635): p. eabm7853.
- Hoepel, W., et al., *High titers and low fucosylation of early human anti-SARS-CoV-2 IgG promote inflammation by alveolar macrophages*. Science Translational Medicine, 2021. **13**(596): p. eabf8654.

While revising the manuscript, we attempted to further highlight and discuss the important work that has been published concerning total and antigen-specific IgG N-glycosylation.

Remarkably, this study shows that this is not reflected in total IgM. However, the lack of specific Ig analysis makes this study less relevant.

We agree that the relevance of this study would be increased if we analyzed the N-glycan profiles from SARS-CoV-2 specific immunoglobulins. Therefore, we analyzed Spike S1-specific IgG and IgM N-glycan profiles from severe and nonsevere COVID-19 cohorts. Please see **Figure 5, and Supplemental Tables 8, 9, and 10** for the analysis of these antigen-specific immunoglobulin N-glycosylation profiles, which support and align with the analysis of total IgG and total IgM N-glycosylation.

Major

-The methods describing glycosyltransferase expression seems to be missing

Methods describing the transcriptomic analysis of glycosyltransferase and glycosidase expression from PBMCs collected from COVID-19 patients in the IMPACC study are listed in the methods section under the subheading: “Nasal viral PCR, host transcriptomics, and metagenomics”. In our revision, we also include a citation to the recent publication in Cell Reports Medicine by the IMPACC group, wherein the methods are described in more detail.

-The rationale to link total IgM glycan features to the amount of IgA/M and G produce to anti-N is absent. What is the hypothesis?

We agree that the rationale in the manuscript was not satisfactorily articulated in the original submission. We hypothesize that the increased S2 and mannose content on IgM correlate with titers of anti-N antibodies – considered markers of disease severity. We have revised the manuscript to indicate the elevation of anti-N immunoglobulins early in COVID-19 are associated with COVID-19 severity, citing work by Vâță et al. 2022 reporting this correlation clinically.

-Epitopes of S1 and RBD are largely overlapping, rationale missing, also why not S2, or why full spike was not used. The reason why S1 and RBD are so different is likely to be a technical problem.

While it is possible that the lower RBD-specific ADCD is due to a technical problem, we cite Butler et al. 2021 (see their figure 2A – Serum ADCD of RBD vs CoV-2 S1) where higher levels of complement were observed in the Spike S1 antigen. Our rationale to examine ADCD with the larger Spike S1 protein is (a) it has been shown that un-vaccinated patients do not develop RBD-specific antibodies early in their disease progression, and (b) Spike S1 offers more antigenic epitopes for antibody binding. We have updated our results section to clearly articulate this point and thank you for the feedback to improve this manuscript.

Furthermore, using plasma also contains IgG which also activates complement. Seems to be compensated in D and E, but conclusion in the results and figure legends are blurry and misleading.

We agree that ADCD detected following incubation with severe and nonsevere cohort plasma is likely driven by IgG. We have updated the results and discussion to better articulate this point. We have also improved the resolution of the figure in the revised manuscript in an attempt to reduce the blurriness of the figure. Our goal was not to mislead when we included plasma in the

ADCD assay, rather we included plasma from severe and nonsevere cohorts in the ADCD assays as a control and a point of reference in comparison to the IgM-dependent ADCD. We have updated the results section to articulate these points in the revised manuscript.

-Discussion is long and not always relevant (and even off topic).

We appreciate your feedback and have reduced the length of the discussion while striving to remain on topic in the revised manuscript.

minor

Results/Figure 1: GP not defined.

Thank you, we have added “GP = glycan peaks” to the **Figure 1A** legend.

Fig. 1B, please show also individual points in B.

Thank you, we have included individual points in the “Sex - % Male” section of **Figure 1B**.

Fig. 1C, the patterns for day 4 and 7 are not clearly distinguishable.

We agree and have changed the Day 7 pattern to large white horizontal lines in an effort to make Day 4 vs Day 7 clearly distinguishable.

Fig 5A middle, it is not clear that you are showing beads here, looks like cells. Please make this clear.

Thank you, we have updated the y-axis label to read “Bead Count” and stressed the fact that these are beads in the figure legend and the results section with the goal of clarifying this point.

Fig. 5A right IgM is shown as a symmetric pentamer by has actually a hexameric configuration with the J chain occupying one of the hexagonal space. It is not clear where the information about the relationship between location and type of glycan in IgM comes from.

Thank you, we apologize for the inaccurate depiction of the IgM pentamer in relation to the J chain. We removed this from the figure. The relationship between the high-mannose site-specific glycosylation comes from sources listed below:

Chandler, K.B., et al., *Multi-isotype Glycoproteomic Characterization of Serum Antibody Heavy Chains Reveals Isotype- and Subclass-Specific N-Glycosylation Profiles*. *Mol Cell Proteomics*, 2019. **18**(4): p. 686-703.

Arnold, J.N., et al., *Human Serum IgM Glycosylation*. *Journal of Biological Chemistry*, 2005. **280**(32): p. 29080-29087.

Loos, A., et al., *Expression and glycoengineering of functionally active heteromultimeric IgM in plants*. *Proceedings of the National Academy of Sciences*, 2014. **111**(17): p. 6263-6268.

We have revised the manuscript to describe IgM site-specific glycosylation more clearly in the introduction, results, and discussion.

Reviewers' Comments:

Reviewer #1:

Remarks to the Author:

Thank you for commenting on the questions raised last time. Overall, my points have been addressed in a satisfactory manner and the authors have improved clarity and rigor of their study.

Reviewer #2:

Remarks to the Author:

5. The authors should show a curve of mannosidase and sialidase-digested IgM to show the specific glycoforms analyzed in Figure 5E – this relates to Figure 2C where the authors show IgM M4-M6 and M7-M10 are significantly regulated oppositely. Which category is more impactful?

Authors' response: This is a great question; it is hard to say which (mannose or sialic acid content) is the more impactful N-glycosylation trend observed on IgM isolated from the severe COVID-19 cohort. The sialidase digestion in Figure 6D suggests sialic acid plays a significant role in IgM complement deposition. We have updated the manuscript to discuss these points.

The author addressed part of the reviewer's question. On the other hand, do the authors have experimental or literature support showing high mannose (M7-M10) and low mannose (M4-M6) regulate immune markers and complement deposition similarly? Specifically, in Figure 4, the authors used 'total mannose' to encapsulate the subtle alterations in M4-M6 and M7-M10 distribution in Figure 2C. It would be recommended to separate 'total mannose' to M4-M6 and M7-M10 and correlate the latter two with immune markers to further elucidate the effect of low- and high-mannose structures on IgM function.

Overall, the authors sufficiently addressed the reviewer's questions – elucidation on Q5 would be beneficial.

Reviewer #3:

Remarks to the Author:

I thank the authors for their comprehensive answers and effort to improve their manuscript. I have only a few points remaining. First regarding the purification of specific immunoglobulins – good job! It might help and be prudent to mention how large these pools were also in the figure legends and materials and methods. Second, it is also important to mention when these samples were taken as previous studies looking at S-specific (also N but to less extent) IgG glycosylation in COVID-19 find afucosylated IgG only transiently after infection. You seem only to find trace amounts of afucosylated IgG, which might be explained by the timing. Please add (I can't find this information).

While I appreciate that you now elaborate more on your hypothesis regarding IgM glycosylation to the amount of specific antibodies produced, I still think this is rather vague and a good rationale lacking and aimed for future testing – nothing wrong with that, except perhaps for the part that your discussion is very long. Which make this study observational – also nothing wrong with that.

REVIEWER COMMENTS

Reviewer #1 (Remarks to the Author):

Haslund-Gourley and colleagues have assessed IgM N-glycosylation patterns in the setting of early acute COVID-19 in a cohort of 22 hospitalized patients, of which 10 had a severe and 12 a non-severe disease course. They found and convincingly show differences in IgM mannose and di-sialylated (S2) N-glycans. Moreover, via a bulk RNA sequencing dataset of the same patients, the authors can link these phenotypes to reduced glycosyltransferase/glycosidase mRNA expression in PBMCs. However, further associations with clinical and serological markers are hindered by the low number of observations, making the interpretation of associated markers such as IL-18 difficult.

The final complement activation assays give more mechanistic insights, as the increased complement activation elicited by purified IgM from severe patients, compared to non-severe patients, was abrogated by the addition of mannosidase or sialidase. This observation further increases the potential relevance of their finding on N-glycosylation patterns of IgM in severe COVID-19 patients. Overall, I found this study well-reported.

Thank you, we appreciate your insight into the limitation of the associations of the IgM N-glycan profile with serological markers and clinical laboratory data. We have revised **Figure 4** to reflect a more conservative approach to the correlations to IgM N-glycans.

Major points:

Figure 3: The glycosyltransferase expression levels appear to be analyzed in all PBMCs and not only B cells/plasmablasts. Can the authors look at only the B cells/ plasmablasts in this dataset?

We would have preferred to analyze B cell/plasmablast-specific glycosyltransferase expression, unfortunately, our collaborators only analyzed total PBMC mRNA expression. We note this important distinction in the results section and elaborated upon this fact in the results and discussion section of the revised manuscript.

Figure 4: The authors strengthen their finding of changes in IgM N-glycosylation and glycosyltransferase expression by correlating the glycosylation patterns with serological correlates of COVID-19 severity. However, the low patient numbers appear to affect the statistical significance, particularly shown by the observation that between the non-severe and the severe patients, only the cytokine IL-18 is elevated as reported in line 549 and Supplementary Table 4. The interpretation that IL-16 and IL-18 “may play a role in controlling downstream glycosyltransferase expression” is questionable based on the presented data. Can the authors provide further evidence for the suggested mechanism? Does for example a linear model correcting for patient age, sex, and severity confirm the association of IL-16/18 with the glycosyltransferase expression levels?

We agree that our interpretation of cytokines IL-16 and IL-18 impacting IgM N-glycosylation does not meet the threshold to claim correlation. We have removed IL-16 and IL-18 from **Figure 4** and updated the results and discussion sections to reflect a more conservative consideration of these cytokines.

Figure 5: The finding of increased complement activation in severe COVID-19 patients by 5ug purified IgM is very interesting. Can differences in the amount of IgM binding to the beads potentially lead to differences in subsequent complement activation, e.g. by different affinities to SARS-CoV-2 spike in the severe, compared to non-severe?

The difference in the amount of IgM binding to the antigen-coated beads could lead to differences in subsequent complement activation. We have updated the results and discussion to address this point in our revised manuscript.

Also, do the authors see correlations between the complement activation capacity and total mannose or S2 on a patient level? Can the abrogation of this effect by mannosidase or sialidase be quantified and correlated to the total mannose or S2 on a patient level?

These are excellent questions, and this would be a fantastic next step for the project, however, at this time we did not have sufficient plasma to explore on an individual level. To complete these studies with the available allocated plasma, we created pools of severe and nonsevere cohorts rather than run individual samples from the five trajectories. Future studies should examine the relationship between S2 and mannose content on an individual patient level. We have updated our future directions to speak to this point.

Minor points:

Table 1: The comparison across the 5 groups is difficult by eye. Can the authors add a table merging the non-severe patients and severe patients respectively (similar to Suppl. Table 4 and most figures of the manuscript).

We have updated Table 1 with two additional columns reflecting the nonsevere or severe cohort grouping to increase the ease of comparison.

Also, please show already in the main table D-Dimer, BUN, and Creatinine measurements as continuous variables with median + interquartile range, instead of as categorical variables, which makes comparisons difficult. Age, BMI, symptom onset to hospitalization days, and SOFA score would also profit from such reporting here.

We have updated Table 1 to present Age, D-Dimer, BUN, Creatinine, and SOFA score as continual variables with mean +/- Standard Deviation (S.D.). However, we feel that BMI and symptom onset to hospitalization days provide more detailed information to the reader remaining in their current format.

Figure 1B: The groups are very well balanced regarding the clinical characteristics, despite having very different COVID-19 severities (one could expect older, and more male patients in group 5). Have the authors matched the trajectory groups selected for this study previous to the experimental exploration?

Patient samples were selected to offer balanced clinical characteristics (sex, BMI, age) prior to experimental assays. Trajectories were determined in the Ozonoff et al. publication. We have revised the human sample methods section to communicate these points.

Were there any inclusion criteria or potential biases for the 22 patients in this study deriving from a cohort with 1164 patients according to the cited paper by Ozonoff et al. ([https://www.thelancet.com/journals/ebiom/article/PIIS2352-3964\(22\)00390-5](https://www.thelancet.com/journals/ebiom/article/PIIS2352-3964(22)00390-5))

These samples were obtained from the Drexel IMPACC plasma biorepository, with the goal of balancing the male and female sexes and the number of patients in the severe vs nonsevere cohorts. There were no other inclusion criteria or potential biases for the 22 patients in this study.

Figure 1C is well presented and convincingly shows the differences within the patient subgroups.

Lines 170-180: Please also consider citing the following key paper by Chakraborty et al. (<https://www.science.org/doi/10.1126/scitranslmed.abm7853>), who additionally show differences in antibody fucosylation between SARS-CoV-2 infected and COVID-19 vaccinated patients.

Thank you, we have updated the manuscript to include this citation on line 203.

Between which months were the patients included in this study? Please report in the methods section.

We have updated the methods section and introduction to inform readers that these sera samples were collected between the months of May 2020 through December 2020.

The lack of vaccination could already mentioned in the methods section on the human subjects.

We have updated the methods section to detail the lack of vaccinations available for the patient populations due to the early stage of the pandemic.

Line 196: The authors introduce the ability of IgM to activate the complement cascade by both the classical and the lectin pathways. Here they cite Hiatt et al., which is a commentary on the engineering of IgM in plants, without mentioning lectin pathway activation neither in the commentary nor the associated original article. The second citation, Zhang et al., shows is based on a natural IgM in a mouse model. Can the authors expand this part of the introduction by also considering data in humans (e.g. [https://www.jbc.org/article/S0021-9258\(20\)56470-3](https://www.jbc.org/article/S0021-9258(20)56470-3))? This could benefit the later interpretation of increased complement activation.

We apologize for the lack of clarity on these citations. We removed the plant IgM engineering commentary article, and instead support the claim of IgM N-glycans interacting with MBL and the lectin complement pathway citing the Arnold et al. 2005 (human), and Zhang et al. 2006 (murine model). Further we cite Sharp et al. 2019 in support of IgM N-glycan's role in classical complement deposition.

This section now reads: *"In addition, evidence supports IgM N-glycans interacting with C1q in the classical complement pathway and the mannan-binding lectin (MBL) associated with the lectin pathway of complement activation."*

Lines 539 -544: Please provide also a direct comparison of SARS-CoV-2-specific antibody levels between the analyzed patient trajectory groups (boxplots).

We recognize the importance of presenting severe and nonsevere patient SARS-CoV-2 specific antibody levels. We present these data in **Supplemental Table 7** and include associated p-values from student's T-test. We have updated the results section to direct interested readers to **Supplemental Table 7** to assess the SARS-CoV-2 specific antibody levels.

Discussion: Do the authors have access to clinical follow-up data on the 22 patients? As total IgM has been associated with long Covid (<https://www.nature.com/articles/s41467-021-27797-1>), it may be interesting to correlate the N-glycosylation with this pattern or discuss the potential relevance of IgM mannose and S2 also in the post-acute setting.

We agree this is an important point to make during the discussion. We have varying degrees of access to clinical follow-up to the patients in this study, but this study was not designed to capture data on long-COVID-19. We have updated our discussion and cite the suggested article in our revised manuscript.

Reviewer #1 (Remarks to the Author): [post-revision remarks]

Thank you for commenting on the questions raised last time. Overall, my points have been addressed in a satisfactory manner and the authors have improved clarity and rigor of their study.

Reviewer #2 (Remarks to the Author):

The study unveils an interesting trend in IgM N-glycosylation in COVID-19 patient cohorts and expands on the current understanding of IgG and IgM on disease states. The study design and analysis are sound, and the presentation is acceptable or adequate. However, the significance of the study is downplayed by the correlational nature of the study, therefore a direct cause-consequence relationship is not clear.

General comments:

1. The bar patterns in Figure 2 are not very differentiating between Day 4 and Day 7. Please use a more distinguishable pattern/color.

Thank you, we have updated **Fig 1C** to better differentiate the Day 4 and Day 7 using large white stripes that more easily visualized.

2. Red dash in Figure 5E figure title should be omitted.

We have omitted the red dash under this figure title (which is now **Figure 6**).

Significant comments:

1. Since the study predominantly focuses on differential glycosylation between Ig serotypes, after IgG / IgM isolation, the authors should assess protein purity and successful purification – e.g. from MS or PAGE.

Thank you, our purification of the total plasma IgG and IgM was confirmed via western blot and SDS-PAGE gels. We have revised the manuscript to include SDS-PAGE Coomassie-stained gels and western blot confirmation of total IgG and total IgM proteins enriched from total plasma in **Supplemental Figures 6A and 6B**.

In addition, we assessed the enrichment of the Spike S1-specific IgG and IgM for LC-MS/MS analysis and include the results of this analysis in **Supplemental Figures 6C and 6D**. We acknowledge there is a low level of IgG detected in the 75kDa band. This does not present a concern as IgM contains 5-times the number of glycosylation sites compared to IgG and reassuringly, the most abundant IgG N-glycan, FA2G0, constitutes only 0.5-0.6% of the IgM N-glycan profile. Thus, any N-glycans from IgG are negligible and do not contribute to the changes we observe within the IgM N-glycan profile.

2. The authors state IgM glycosylation is more impactful on immune regulation. However, if the trend in sialylation is opposite in trends to IgG, how do the authors rule out whether the trends in immunogenicity are due to IgG or IgM – particularly because the immunogenicity trends the authors mention correlate to established sialylation characteristics in IgG?

We do not intend to claim that IgM N-glycosylation is more impactful on immune regulation compared to the role that IgG N-glycans play. Rather, we seek to highlight the potential roles of IgM N-glycans during severe COVID-19. We have revised the manuscript to clarify this point.

Specifically, in Figures 2 and 3, the authors show glycosyltransferase expression differs in the two trajectories (1-3, 4-5), whereas only IgM glycosylation shows a significant difference. The authors should also show a similar categorization by disease trajectory for Figure 4 to justify their statement about IgM glycosylation.

This is an important point, we have included **Supplementary Tables 4, 5, and 6** in the revised manuscript to assess IgG agalactosylation, di-galactosylation, and mono-sialylation correlations with the glycosyltransferase and glycosidase transcriptomic dataset.

3. Related to Q2, outside of correlational data, the study shows direct evidence of the immunogenic response of IgM only in the last figure. The argument for the glycosylation/sialylation effect resulting for IgM would have been stronger if binding/activity assays would be implemented with purified IgM.

We agree that data connecting the glycosylation of IgM to the immunogenic response of IgM-dependent ADCD required further experimentation. During our revisions, we isolated SARS-CoV-2 Spike S1-specific IgG and IgM from severe and nonsevere COVID-19 cohorts (**Figure 5**). This analysis revealed that Spike S1-specific IgG N-glycosylation presented with a highly pro-inflammatory glycan phenotype, with elevated levels of FA2G0 in the severe COVID-19 cohort. On the other hand, Spike S1-specific IgM N-glycosylation contained elevated levels of sialic acid and high-mannose (M7-M10) moieties in the severe COVID-19 cohort compared to the nonsevere cohort. We believe that the antigen-specific IgM N-glycosylation analysis better supports the observed differences between the severe and nonsevere IgM-dependent ADCD presented in **Figure 6**.

4. In the manuscript abstract figure, the authors hypothesized IL-18 and IL-16 might alter glycosyltransferase expression levels. The figure shows a question mark indicating the entire picture is not complete. Alternatively, can the authors show a linkage between IL-18/IL-16 addition/induction and glycosyltransferase expression in a model cell line?

We thank the reviewer for this comment and agree that we do not present enough evidence in support of IL-18 and IL-16 to directly impact the glycosylation of IgM. We revised our manuscript to downplay the potential role of IL-18 and IL-16.

5. The authors should show a curve of mannosidase and sialidase-digested IgM to show the specific glycoforms analyzed in Figure 5E – this relates to Figure 2C where the authors show IgM M4-M6 and M7-M10 are significantly regulated oppositely. Which category is more impactful?

This is a great question; it is hard to say which (mannose or sialic acid content) is the more impactful N-glycosylation trend observed on IgM isolated from the severe COVID-19 cohort. The

sialidase digestion in **Figure 6D** suggests sialic acid plays a role in IgM complement deposition. We have updated the manuscript to discuss these points.

6. What is the author's interpretation of Figure 5C-E? In Figure 5E, it appears IgM is not responsive to complement deposition in phases 1-3. Therefore, the complement activation response in plasma is likely from other plasma components or Ig serotypes – which is consistently observed in states 1-3 and 4-5. On the other hand, with severe disease states (4-5), sialidase treatment had a strong effect on IgM complement deposition but did not seem to be responsive in complement activation in plasma. Don't these observations indicate that N-glycosylation of IgM is not responsible for complement deposition in plasma, but other components/serotypes? Please clarify in manuscript text.

Thank you. We agree that in total plasma, the deposition of complement in an antigen-specific manner is predominantly promoted by IgG. This is because IgG is highly abundant and may have a higher affinity to the presented antigen. We have clarified this point in the revised manuscript.

Nevertheless, purified IgM also induces complement deposition in an antigen-specific manner, and that exoglycosidase digestions of the purified IgM modulate the ability of Spike S1-specific IgM to deposit complement. This suggests that IgM contributes to the overall rate of complement deposition in COVID-19 patients and may promote more complement deposition in the severe COVID-19 cohort. We have revised the manuscript to clarify these points as well.

7. Related to Q6, did the authors measure the percentage of IgM serotypes compared to total Ig?

During our revision, we isolated Spike S1-specific immunoglobulins from severe and nonsevere COVID-19 cohorts and analyzed the resulting abundance of the Spike S1-specific IgG (50kDa), and IgM (75kDa) using a Coomassie-stained SDS-PAGE gel followed by LC-MS/MS presented in **Supplementary Figures 6C and 6D**. There, we observed a higher abundance of the IgM heavy chain in the severe COVID-19 cohort compared to the nonsevere cohort. We have revised the manuscript to include this finding.

8. Related to Q2&6, the authors should show IgG N-glycosylation correlation statistics as a supplementary figure or table to show the significance of IgM, and not IgG, glycosylation on clinical and transcriptomic parameters.

This is an excellent point. We revised the manuscript to include **Supplementary Tables 4, 5, and 6** which compares the IgG N-glycosylation that significantly differed between the severe and nonsevere cohorts (G0, G2, and S1). We have cited this table in the results section to contrast the findings of IgM to those of IgG N-glycosylation. In brief, the IgG N-glycosylation data did not correlate with BUN, creatinine, potassium, or anti-N antibody abundance. However, IgG and IgM N-glycosylation trends correlated with elevated D-dimer and decreased MAN1A2 expression in

the severe cohort. We have revised the manuscript to highlight these results in the results and discussion sections.

Reviewer #2 (Remarks to the Author): [post-revision remarks]

5. The authors should show a curve of mannosidase and sialidase-digested IgM to show the specific glycoforms analyzed in Figure 5E – this relates to Figure 2C where the authors show IgM M4-M6 and M7-M10 are significantly regulated oppositely. Which category is more impactful?

Authors' response: This is a great question; it is hard to say which (mannose or sialic acid content) is the more impactful N-glycosylation trend observed on IgM isolated from the severe COVID-19 cohort. The sialidase digestion in Figure 6D suggests sialic acid plays a role in IgM complement deposition. We have updated the manuscript to discuss these points.

5.1 The author addressed part of the reviewer's question. On the other hand, do the authors have experimental or literature support showing high mannose (M7-M10) and low mannose (M4-M6) regulate immune markers and complement deposition similarly?

IgM N-glycans have not been studied to the extent of IgG. We were unable to find literature documenting high mannose (M7-M10) and low mannose (M4-M6) regulating immune markers and complement deposition. As a result of this work, we plan to continue our investigation into the effect of aberrant IgM mannosylation on the immune system.

Specifically, in Figure 4, the authors used 'total mannose' to encapsulate the subtle alterations in M4-M6 and M7-M10 distribution in Figure 2C. It would be recommended to separate 'total mannose' to M4-M6 and M7-M10 and correlate the latter two with immune markers to further elucidate the effect of low- and high-mannose structures on IgM function.

Overall, the authors sufficiently addressed the reviewer's questions – elucidation on Q5 would be beneficial.

We appreciate your suggestion to separate M4-M6 and M7-M10 and correlate each population to patient clinical markers of disease severity. We see the merit in breaking down the changes observed between the smaller and larger mannose species. However, we believe that Figure 2C provides insight into the shift toward less processed mannose structures which is further supported in Figures 3A and 3B. On the other hand, the linear correlation plots in Figure 4 highlight the importance of changes in overall IgM mannosylation in relation to clinical outcomes, similar to the changes in total S2 content on IgM.

Reviewer #3 (Remarks to the Author):

The paper of Haslund-Gourley et al describes the changes in IgM (and G) total galactosylation in the first few weeks after a naïve SARS-CoV-2 infection. While interesting, to my knowledge IgM has not been analyzed in COVID-19 before, the study suffers tremendously from previous publication of specific IgG glycosylation. The relevance looking only at specific IgG glycosylation is perhaps best exemplified in figure 2, where the most severe individuals show lowered total IgG fucosylation, that is reminiscent to what has been published before, and is most likely the result of the massive amount of specific IgG produced that is transiently afucosylated. The changes in IgG galactosylation in the most severe individuals has also been described before (very strong reduction of both specific (from 80 to 20% but also total IgG galactosylation seen in the most severe). None of the studies looking at antigen-specific (and total) IgG glycosylation in this infection is introduced as context nor discussed.

We acknowledge the reviewer's important point about examining the antigen-specific immunoglobulin glycosylation, However, we respectfully disagree with their assertion that "None of the studies looking at antigen-specific (and total) IgG glycosylation in this infection is introduced as context nor discussed."

In the introduction of our manuscript, we cite and discuss sources (below) demonstrating previous IgG N-glycosylation studies during severe COVID-19:

- Pongracz, T., et al., *Immunoglobulin G1 Fc glycosylation as an early hallmark of severe COVID-19*. EBioMedicine, 2022. **78**: p. 103957.
- Hou, H., et al., *Profile of Immunoglobulin G N-Glycome in COVID-19 Patients: A Case-Control Study*. Frontiers in Immunology, 2021. **12**.
- Kljaković-Gašpić Batinjan, M., et al., *Differences in Immunoglobulin G Glycosylation Between Influenza and COVID-19 Patients*. Engineering (Beijing), 2022.
- Petrović, T., et al., *IgG N-glycome changes during the course of severe COVID-19: An observational study*. eBioMedicine, 2022. **81**: p. 104101.
- Vicente, M.M., et al., *Altered IgG glycosylation at COVID-19 diagnosis predicts disease severity*. European Journal of Immunology, 2022. **52**(6): p. 946-957.
- Chakraborty, S., et al., *Early non-neutralizing, afucosylated antibody responses are associated with COVID-19 severity*. Science Translational Medicine, 2022. **14**(635): p. eabm7853.
- Hoepel, W., et al., *High titers and low fucosylation of early human anti-SARS-CoV-2 IgG promote inflammation by alveolar macrophages*. Science Translational Medicine, 2021. **13**(596): p. eabf8654.

While revising the manuscript, we attempted to further highlight and discuss the important work that has been published concerning total and antigen-specific IgG N-glycosylation.

Remarkably, this study shows that this is not reflected in total IgM. However, the lack of specific Ig analysis makes this study less relevant.

We agree that the relevance of this study would be increased if we analyzed the N-glycan profiles from SARS-CoV-2 specific immunoglobulins. Therefore, we analyzed Spike S1-specific IgG and IgM N-glycan profiles from severe and nonsevere COVID-19 cohorts. Please see **Figure 5, and Supplemental Tables 8, 9, and 10** for the analysis of these antigen-specific immunoglobulin N-glycosylation profiles, which support and align with the analysis of total IgG and total IgM N-glycosylation.

Major

-The methods describing glycosyltransferase expression seems to be missing

Methods describing the transcriptomic analysis of glycosyltransferase and glycosidase expression from PBMCs collected from COVID-19 patients in the IMPACC study are listed in the methods section under the subheading: “Nasal viral PCR, host transcriptomics, and metagenomics”. In our revision, we also include a citation to the recent publication in Cell Reports Medicine by the IMPACC group, wherein the methods are described in more detail.

-The rationale to link total IgM glycan features to the amount of IgA/M and G produce to anti-N is absent. What is the hypothesis?

We agree that the rationale in the manuscript was not satisfactorily articulated in the original submission. We hypothesize that the increased S2 and mannose content on IgM correlate with titers of anti-N antibodies – considered markers of disease severity. We have revised the manuscript to indicate the elevation of anti-N immunoglobulins early in COVID-19 are associated with COVID-19 severity, citing work by Vâță et al. 2022 reporting this correlation clinically.

-Epitopes of S1 and RBD are largely overlapping, rationale missing, also why not S2, or why full spike was not used. The reason why S1 and RBD are so different is likely to be a technical problem.

While it is possible that the lower RBD-specific ADCD is due to a technical problem, we cite Butler et al. 2021 (see their figure 2A – Serum ADCD of RBD vs CoV-2 S1) where higher levels of complement were observed in the Spike S1 antigen. Our rationale to examine ADCD with the larger Spike S1 protein is (a) it has been shown that un-vaccinated patients do not develop RBD-specific antibodies early in their disease progression, and (b) Spike S1 offers more antigenic epitopes for antibody binding. We have updated our results section to clearly articulate this point and thank you for the feedback to improve this manuscript.

Furthermore, using plasma also contains IgG which also activates complement. Seems to be compensated in D and E, but conclusion in the results and figure legends are blurry and misleading.

We agree that ADCD detected following incubation with severe and nonsevere cohort plasma is likely driven by IgG. We have updated the results and discussion to better articulate this point. We have also improved the resolution of the figure in the revised manuscript in an attempt to reduce the blurriness of the figure. Our goal was not to mislead when we included plasma in the

ADCD assay, rather we included plasma from severe and nonsevere cohorts in the ADCD assays as a control and a point of reference in comparison to the IgM-dependent ADCD. We have updated the results section to articulate these points in the revised manuscript.

-Discussion is long and not always relevant (and even off topic).

We appreciate your feedback and have reduced the length of the discussion while striving to remain on topic in the revised manuscript.

minor

Results/Figure 1: GP not defined.

Thank you, we have added “GP = glycan peaks” to the **Figure 1A** legend.

Fig. 1B, please show also individual points in B.

Thank you, we have included individual points in the “Sex - % Male” section of **Figure 1B**.

Fig. 1C, the patterns for day 4 and 7 are not clearly distinguishable.

We agree and have changed the Day 7 pattern to large white horizontal lines in an effort to make Day 4 vs Day 7 clearly distinguishable.

Fig 5A middle, it is not clear that you are showing beads here, looks like cells. Please make this clear.

Thank you, we have updated the y-axis label to read “Bead Count” and stressed the fact that these are beads in the figure legend and the results section with the goal of clarifying this point.

Fig. 5A right IgM is shown as a symmetric pentamer by has actually a hexameric configuration with the J chain occupying one of the hexagonal space. It is not clear where the information about the relationship between location and type of glycan in IgM comes from.

Thank you, we apologize for the inaccurate depiction of the IgM pentamer in relation to the J chain. We removed this from the figure. The relationship between the high-mannose site-specific glycosylation comes from sources listed below:

Chandler, K.B., et al., *Multi-isotype Glycoproteomic Characterization of Serum Antibody Heavy Chains Reveals Isotype- and Subclass-Specific N-Glycosylation Profiles*. *Mol Cell Proteomics*, 2019. **18**(4): p. 686-703.

Arnold, J.N., et al., *Human Serum IgM Glycosylation*. *Journal of Biological Chemistry*, 2005. **280**(32): p. 29080-29087.

Loos, A., et al., *Expression and glycoengineering of functionally active heteromultimeric IgM in plants*. Proceedings of the National Academy of Sciences, 2014. **111**(17): p. 6263-6268.

We have revised the manuscript to describe IgM site-specific glycosylation more clearly in the introduction, results, and discussion.

Reviewer #3 (Remarks to the Author): [post-revision remarks]

I thank the authors for their comprehensive answers and effort to improve their manuscript. I have only a few points remaining.

First regarding the purification of specific immunoglobulins – good job! It might help and be prudent to mention how large these pools were also in the figure legends and materials and methods. Second, it is also important to mention when these samples were taken as previous studies looking at S-specific (also N but to less extent) IgG glycosylation in COVID-19 find afucosylated IgG only transiently after infection. You seem only to find trace amounts of afucosylated IgG, which might be explained by the timing. Please add (I can't find this information).

Thank you, we have updated the materials and methods to specify the size of the pools used to obtain Spike S1-specific immunoglobulin N-glycan profiles and that these plasma samples were collected from day 4 of the patient's COVID-19 hospitalization. Further, we updated the figure legend to specify these plasma samples were collected from consenting patients on day 4 of hospitalization.

While I appreciate that you now elaborate more on your hypothesis regarding IgM glycosylation to the amount of specific antibodies produced, I still think this is rather vague and a good rationale lacking and aimed for future testing – nothing wrong with that, except perhaps for the part that your discussion is very long. Which make this study observational – also nothing wrong with that.

Thank you, we attempted to reduce the length of the discussion during the revision. As this manuscript is the first to report changes in IgM N-glycosylation during human viral infection, we aimed to comprehensively address our findings and cite the relevant literature to contextualize our findings within the discussion.